# Current-induced manipulation of exchange bias in IrMn/NiFe bilayer structures

Jaimin Kang [1], Jeongchun Ryu[1✉], Jong-Guk Choi[1], Taekhyeon Lee[2], Jaehyeon Park[2], Soogil Lee [1], Hanhwi Jang[1], Yeon Sik Jung [1], Kab-Jin Kim [2] & Byong-Guk Park [1✉]

The electrical control of antiferromagnetic moments is a key technological goal of antiferromagnet-based spintronics, which promises favourable device characteristics such as ultrafast operation and high-density integration as compared to conventional ferromagnet-based devices. To date, the manipulation of antiferromagnetic moments by electric current has been demonstrated in epitaxial antiferromagnets with broken inversion symmetry or antiferromagnets interfaced with a heavy metal, in which spin-orbit torque (SOT) drives the antiferromagnetic domain wall. Here, we report current-induced manipulation of the exchange bias in IrMn/NiFe bilayers without a heavy metal. We show that the direction of the exchange bias is gradually modulated up to ±22 degrees by an in-plane current, which is independent of the NiFe thickness. This suggests that spin currents arising in the IrMn layer exert SOTs on uncompensated antiferromagnetic moments at the interface which then rotate the antiferromagnetic moments. Furthermore, the memristive features are preserved in sub-micron devices, facilitating nanoscale multi-level antiferromagnetic spintronic devices.

[1] Department of Materials Science and Engineering and KI for Nanocentury, KAIST, Daejeon 34141, Korea. [2] Department of Physics, KAIST, Daejeon 34141, Korea. ✉email: jeongchun330@gmail.com; bgpark@kaist.ac.kr

Antiferromagnets (AFM), magnetically ordered materials with neighboring magnetic moments pointing in opposite direction, exhibit the absence of macroscopic magnetization and are robust against external magnetic fields. This, together with their terahertz spin dynamics, promises the development of AFM-based spintronic devices with high-density integration and ultrafast operation that can transcend the capabilities of existing ferromagnet-based spintronic devices[1–8]. On the other hand, the negligible net magnetization makes it difficult to control AFM moments with magnetic fields. Therefore, finding efficient techniques to manipulate AFM order, preferably by electrical means, is of critical importance to realize AFM-based spintronic devices.

The electrical manipulation of AFM moments has been demonstrated in previous reports and can be divided into two categories. The first is to employ a single AFM layer with spatial broken inversion symmetry such as CuMnAs[9–11] or Mn$_2$Au[12–15]. In these materials, electric currents locally induce non-equilibrium spin polarization, generating Néel spin–orbit torque (SOT) with opposite signs for each sub-lattice with opposite magnetic moments[16]. However, the Néel SOT demands a specific crystal symmetry, in which magnetic atoms occupy non-centrosymmetric lattices to form space inversion partners; thus, highly ordered films are required[9–18]. The second is to use AFM/heavy metal bilayers, in which the AFM moment is controlled by the spin current generated by the spin Hall effect[19] in the heavy metal layer and the Rashba–Edelstein effect (REE)[20,21] at the interface. This is similar to the intensively investigated SOT in ferromagnet/heavy metal bilayers[22–25]. A wide range of AFMs has been explored in bilayer structures, including metallic AFMs such as IrMn[26–30], PtMn[31,32], and MnN[33], an insulating AFM, NiO[34–37], and a Weyl semimetal AFM, Mn$_3$Sn[38].

Notably, it has been demonstrated that the electrical manipulation of AFM moments typically shows multi-level characteristics;[17,31,39] the direction of the AFM moment can be gradually modulated by the magnitude and polarity of a writing current. However, the multi-level characteristics rely on the AFM domain structure because the current-induced SOT controls the overall AFM moments by switching the AFM moment in some domains and/or by driving the AFM domain wall[10,28,32,39]. Therefore, to exploit this memristive behavior in nano-devices, it is necessary to either engineer an AFM with nanometer-sized domains or find a way to control the entire AFM moment collectively.

In this article, we report the current-induced manipulation of the exchange bias in IrMn/NiFe bilayer structures. We observe that the planar Hall resistance of the bilayer is gradually modulated by an in-plane current and retains its value even after turning off the current. This demonstrates that the SOT caused by the spin Hall effect in IrMn effectively controls the exchange bias direction in a range of ±22°. To understand the switching mechanism, we investigate the dependence of the rotation angle of the exchange bias ($\varphi_{EB}$) on the IrMn and NiFe thicknesses; $\varphi_{EB}$ diminishes with an increase in the IrMn thickness, indicating that the SOT-induced rotation of the exchange bias is hindered by the AFM anisotropy energy, which increases with its thickness. Interestingly, $\varphi_{EB}$ remains constant regardless of the NiFe thickness up to 10 nm. This implies that the SOT is not applied directly to the ferromagnetic NiFe layer. Therefore, the SOT seems to be applied to the uncompensated AFM moments at the IrMn/NiFe interface, subsequently triggering the collective rotation of the magnetization of the exchange-coupled IrMn/NiFe bilayers. Furthermore, we show that the reversible memristive features of the SOT-induced AFM switching are maintained in a 500-nm-sized device, offering a route for developing nanoscale AFM spintronics devices for applications in neuromorphic computing.

## Results

### Current-induced manipulation of the exchange bias in IrMn/NiFe.

To demonstrate electrical control of the exchange bias, we employ IrMn/NiFe exchange-biased bilayers. In such structures, a charge current induces a spin current through the spin Hall effect in the IrMn layer, exerting torques on the magnetization of the exchange-coupled IrMn/NiFe bilayer. Note that IrMn, a widely used AFM material for exchange bias, exhibits a sizeable (inverse) spin Hall effect[40–44]. As schematically illustrated in Fig. 1a, when applying a charge current in the $x$-direction, a spin current flowing in the $z$-direction has spin polarization $\sigma$ in the $y$-direction, thus pushing the AFM/FM magnetic moment toward the $y$-direction, parallel to $\sigma$. We fabricate a (111) textured polycrystalline IrMn (5 nm)/NiFe (4 nm) bilayer by sputtering deposition on high-resistive Si (675 μm) substrate (Supplementary Note 1) and subsequently anneal it under a magnetic field along the $x$-direction to induce a unidirectional exchange-bias field ($B_{EB}$). Figure 1b shows the magnetization curves of the sample while sweeping the magnetic fields in the $x$- (blue squares) and $y$- (red circles) directions. The hysteresis loop shifts toward the negative field direction only for the measurement along the $x$-direction, demonstrating the exchange bias developed in the IrMn (5 nm)/NiFe (4 nm) bilayer along the positive $x$-direction. The samples are then patterned into a 4-μm-wide Hall bar structure for electrical measurements. First, we measure the planar Hall resistance ($R_H$) of the IrMn/NiFe bilayer while rotating the sample in the $x$–$y$ plane under a magnetic field of 100 mT, which is sufficient to saturate the magnetization. Figure 1c shows the $R_H$ as a function of the azimuthal angle of the magnetic field $\varphi_B$, which allows us to extract the magnetization direction of the IrMn/NiFe bilayer $\varphi_m$ from the measured $R_H$ value.

We next present the main result of this work; the manipulation of the exchange bias through the in-plane current-induced SOT in the IrMn/NiFe structure. Figure 1d shows the changes in $R_H$ of the IrMn/NiFe bilayer as a function of the in-plane current pulse $J_P$ with a width of 30 μs. For each $J_P$, $R_H$ is measured with a reading current of 100 μA after applying $J_P$. Initially, the magnetization direction is in the $x$-direction ($\varphi_m = 0°$), and the corresponding $R_H$ value is set to zero by removing an offset. As $J_P$ increases positively (solid red symbols), $R_H$ remains unchanged until $J_P = 4.2 \times 10^{11}$ A/m² and gradually increases when $J_P$ exceeds $4.2 \times 10^{11}$ A/m². Finally, for $J_P = \sim 8.1 \times 10^{11}$ A/m², the $R_H$ value saturates to $-0.14$ Ω, which corresponds to $\varphi_m = -15°$. We observe similar behavior of the $R_H$ and the corresponding $\varphi_m$, but opposite signs when a negative $J_P$ is applied (solid blue symbols). This demonstrates that the magnetization of the IrMn/NiFe structure is rotated clockwise (counterclockwise) by a positive (negative) in-plane current. Moreover, when we sweep $J_P$ between $\pm 9.2 \times 10^{11}$ A/m² (open red/blue symbols), the $R_H$ value varies between $\mp 0.14$ Ω, demonstrating the repeatable commutation of $\varphi_m$ between $\mp 15°$ with electric current. Note that the current density is kept to below $1.0 \times 10^{12}$ A/m² to prevent the breakdown of the device. We also measure the AMR effect of the bilayer, whose hysteresis loop shifts according to the sign of $\varphi_{EB}$ (Supplementary Note 2). This is consistent with the PHE results, demonstrating that the current-induced rotation of exchange bias occurs over the entire sample.

We further investigate whether the change in the $R_H$ value reflects the rotation of the exchange bias of IrMn, given that the $R_H$ value of the IrMn/NiFe bilayer is mostly dominated by NiFe (Supplementary Note 3). To this end, we measure the dependence of $R_H$ on the magnetic field along the $x$-direction $B_x$. Prior to the measurement, we apply an $J_P$ of $8.3 \times 10^{11}$ A/m² to set $R_H = 0.14$ Ω (or $\varphi_m = +15°$). Figure 1e shows that $R_H$ gradually decreases with an increase in $B_x$ (red squares) and approaches zero when $B_x = 200$ mT. This indicates that the magnetization rotates

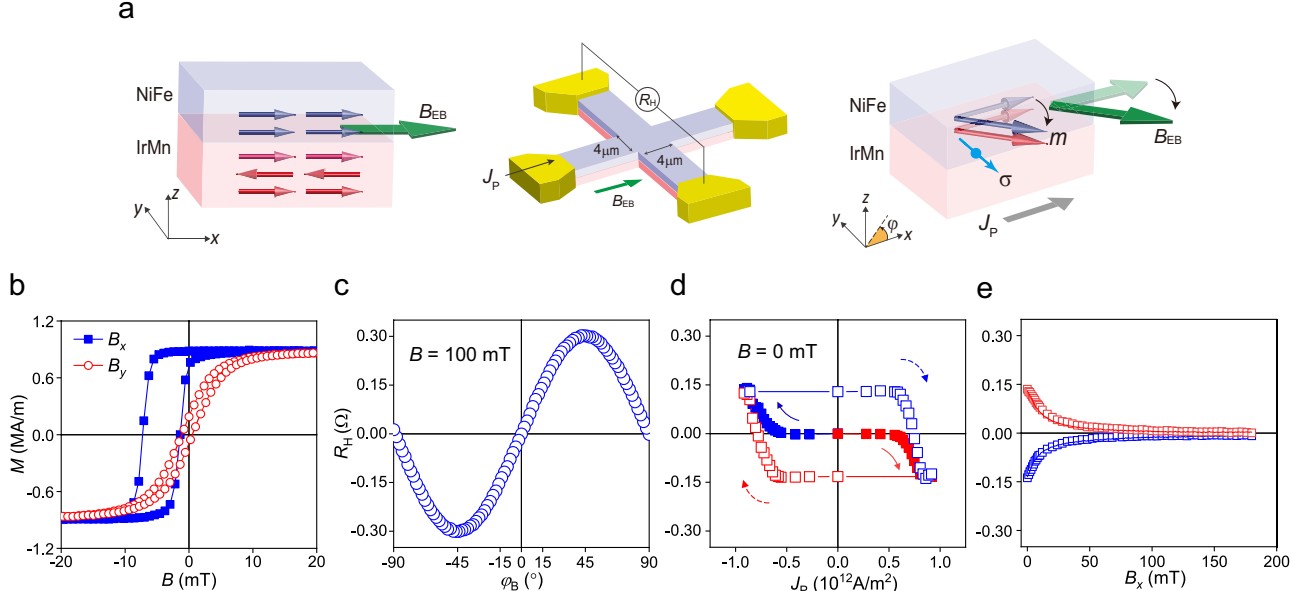

**Fig. 1 Current-induced manipulation of the exchange bias in an IrMn/NiFe structure. a** Left: Exchange-biased IrMn/NiFe structure, where the blue and red arrows represent the magnetization ($m$) of NiFe and IrMn, respectively, and $B_{EB}$ is the exchange bias field. Middle: The Hall bar device used for electrical measurements. The Hall resistance ($R_H$) is measured after applying an in-plane current pulse $J_P$. Right: Schematics of the current-induced exchange bias switching, where the magnetization direction of the IrMn/NiFe bilayer ($\varphi_m$) is modulated by the spin current with spin polarization ($\sigma$). **b** Hysteresis loop of IrMn (5 nm)/NiFe (4 nm) structure measured with magnetic fields along the $x$-axis (solid blue) and $y$-axis (open red). **c** $R_H$ versus azimuthal angle of a magnetic field ($\varphi_B$) of 100 mT. **d** The $R_H$ versus $J_P$ curves, where the arrows denote the sweeping direction of $J_P$. **e** $R_H$ as a function of a magnetic field along $x$-axis ($B_x$). $R_H$ is initially set to $\pm 0.14$ by an $J_P$ of $\mp 8.3 \times 10^{11}$ A/m$^2$, represented by the red and blue symbols. Open squares (lines) refer to increasing (decreasing) $B_x$. The inset illustrates the magnetization changes by $B_x$.

toward the magnetic field direction as expected. Interestingly, when reducing $B_x$, $R_H$ is restored to its initial value (red line in Fig. 1e); i.e., the magnetization rotates back to $+15°$, as depicted in the inset of Fig. 1e. The same behavior is observed when $R_H$ is initialized to $-0.14\,\Omega$ (blue squares and line in Fig. 1e). The recovery of $R_H$ (or $\varphi_m$) indicates that there is a bias field acting on NiFe in the direction of $\varphi = \pm 15°$, which we attribute to the exchange bias originating from the AFM IrMn. This result provides evidence that the exchange bias direction $\varphi_{EB}$ ($//\varphi_m$) is electrically manipulated in an IrMn/NiFe bilayer structure via the current-induced SOT. Similar switching behavior is also observed in other IrMn/FM structures with various FMs of CoFe, CoFeB, and Ni (Supplementary Note 4) demonstrating that the SOT-induced manipulation of exchange bias occurs generally in IrMn/FM exchange-biased structures. On the other hand, this phenomenon is absent in a Ta/NiFe sample, in which spin current is effectively generated by the spin Hall effect in Ta[45,46]. In a structure without an AFM layer, the magnetization rotated by the SOT returns to its easy axis (that is in the $x$-direction) once the current is turned off, resulting in no variation of $R_H$ and $\varphi_m$ (Supplementary Note 5). These results confirm that the AFM IrMn plays a critical role in the electrical modulation of $\varphi_m$ in the IrMn/NiFe structure.

**Current-induced thermal effects.** We note that the exchange bias direction can be rotated by the current-induced thermal annealing effect if the applied current increases the sample temperature above the blocking temperature of the IrMn/NiFe layer and creates an Oersted field larger than the coercivity of the NiFe. To rule out the thermal effects as a possible cause, we first estimate the sample temperature due to the current application by measuring the longitudinal resistance $R_{xx}$ of the IrMn (5 nm)/NiFe (4 nm) sample. Figure 2a shows that $R_{xx}$ increases quadratically with current density $J_P$ ranging from $1.25 \times 10^{11}$ to $8.4 \times 10^{11}$ A/m$^2$.

Then, we measure $R_{xx}$ as a function of temperature (Fig. 2b). By comparing those two results shown in Fig. 2a, b, we estimate the sample temperature ($T_{sample}$). Figure 2c shows that $T_{sample}$ increases to 354 K when applying a critical switching current of $7.4 \times 10^{11}$ A/m$^2$. The mild temperature increase in our sample is due to a large thermal conductivity of the high-resistive Si substrate (Supplementary Note 6). Next, we measure the blocking temperature of the IrMn (5 nm)/NiFe (4 nm) sample. Figure 2d shows magnetization curves measured at various temperatures ranging from 100 to 385 K. The exchange bias field ($B_{EB}$) determined by the hysteresis shift disappears at temperatures greater than 385 K (Fig. 2e and Supplementary Note 7), which is the blocking temperature of our sample. This demonstrates that the temperature of the sample in our experiments is lower than the blocking temperature and Joule heating cannot be the main cause of the observed effect. Furthermore, we observed that a similar switching behavior is obtained when the experiment is performed at low temperatures down to 100 K, much lower than the blocking temperature (Supplementary Note 8).

We subsequently check if the Oersted field due to the injecting current can cause the current-induced modulation of the exchange bias. To this end, we first calculated the Oersted field acting on the NiFe layer in the IrMn/NiFe sample (Supplementary Note 9), which is ~0.68 mT at a switching current density of $7.4 \times 10^{11}$ A/ m$^2$. Then, we measure planar Hall resistance $R_H$ of the IrMn (5 nm)/NiFe (4 nm) sample at 354 K while rotating an external magnetic field of 1 mT. Figure 2f shows that the $R_H$ measured with 1 mT remains almost constant compared to that measured with 200 mT. This demonstrates that such a small Oersted field of less than 1 mT cannot make any noticeable change in the magnetization direction of the exchanged coupled IrMn/NiFe sample. These results corroborate that current-induced thermal effects, including Joule heating and Oersted field, are not the primary cause of the current-induced manipulation of exchange bias.

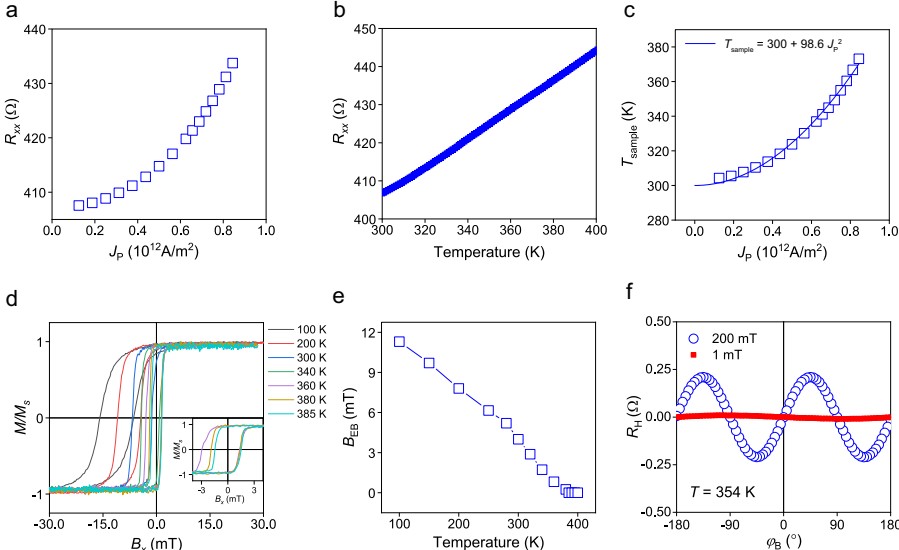

**Fig. 2 Current-induced thermal effects in the IrMn (5 nm)/NiFe (4 nm) structure. a** Longitudinal resistance $R_{xx}$ as a function of current density $J_P$ ranging from $1.25 \times 10^{11}$ to $8.4 \times 10^{11}$ A/m². **b** $R_{xx}$ measured as a function of temperature. **c** Estimated sample temperature $T_{sample}$ as a function of current density $J_P$. The quadratic line represents a fitting curve of $T = 300 + 98.6 J_P^2$. **d** Normalized magnetization loops at various temperatures between 100 and 385 K. Inset shows magnified magnetization loops measured at 360–385 K. **e** Exchange bias field $B_{EB}$ versus temperature. **f** Planar Hall resistance $R_H$ measured with external magnetic fields of 200 and 1 mT at 354 K.

**Mechanisms of SOT-induced switching of exchange bias.** To understand the underlying mechanism of the current-induced manipulation of the exchange bias, we investigate various IrMn/NiFe structures. Assuming that the spin current is only generated in the IrMn layer, there are two possible scenarios; first, the spin current exerts torques on the AFM moments itself. Second, the spin current induces spin accumulation on the IrMn/NiFe interface, giving torques to NiFe and controlling its magnetization. This is followed by the rotation of the exchange-coupled IrMn moment. In the latter case, where opposite spins are accumulated on the top and bottom interfaces, the rotation direction will reverse when changing the stacking order, whereas it is independent of the stacking order in the former case. To verify this, we measure the $R_H$ versus $J_P$ curves using a NiFe (4 nm)/IrMn (5 nm) sample, an inversion of the IrMn/NiFe structure shown in Fig. 1d. Figure 3a shows that the $\varphi_{EB}$ extracted from the $R_H$ value rotates reversibly within ±15° by the in-plane current, but its polarity is reversed. This indicates that the spin current in the IrMn and associated spin accumulation at the IrMn/NiFe interface are primarily responsible for the current-induced manipulation of the magnetization of IrMn/NiFe bilayers. Furthermore, we examine an IrMn (5 nm)/NiFe (4 nm)/Ta (1.5 nm) structure, where the Ta layer diminishes the Oersted field effect (Supplementary Note 9), but it provides additional spin currents injected into the NiFe layer since Ta has a negative spin Hall angle opposite to IrMn[40,43] (Supplementary Note 10). Figure 3b shows the results; by introducing the Ta layer, the current-induced variation of $\varphi_{EB}$ is enhanced to ±22° while switching polarity remains the same. This again indicates that the Oersted field contribution is not significant in this measurement, accentuating that the spin current in IrMn and the associated SOT is the main cause of the current-induced manipulation of the AFM moment.

We want to discuss other possible contributions that can generate spin torques to the current-induced manipulation of exchange bias direction, such as the REE of the IrMn/NiFe interface or the spin anomalous Hall effect (SAHE) of the NiFe layer. First, we cannot completely exclude the REE originating from the IrMn/NiFe interface due to the same symmetry as the bulk spin Hall effect. However, field-like SOT, governed primarily

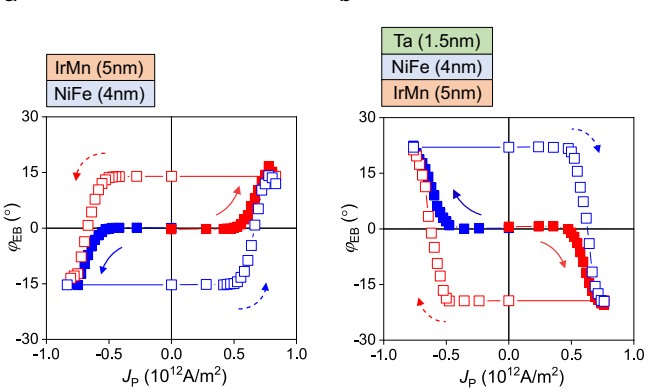

**Fig. 3 SOT-induced exchange bias switching for various IrMn/NiFe structures. a**, **b** Direction of the exchange bias ($\varphi_{EB}$) versus $J_P$ curves for the NiFe (4 nm)/IrMn (5 nm) sample (**a**) and the IrMn (5 nm)/NiFe (4 nm)/Ta (1.5 nm) sample (**b**). The arrows denote the sweeping direction of $J_P$. The $\varphi_{EB}$ values are extracted from the $R_H$ values.

by the REE, is very small in our samples (Fig. S11), indicating less contribution from the REE effect than the spin Hall effect. Second, the SAHE can also generate a spin current, of which spin polarization is parallel to the magnetization direction[47]. For the SAHE generated from NiFe to exert torques on the uncompensated moments of IrMn, the magnetization direction of NiFe must be different from the direction of the IrMn uncompensated moments. However, it is not the case; the exchange bias of the IrMn/NiFe structures manifests that the two directions are aligned in parallel to each other. Therefore, we exclude the SAHE as a possible cause.

We subsequently study the thickness dependence of the SOT-induced manipulation of the exchange bias. Figure 4a shows the hysteresis loops of the IrMn ($t_{IrMn}$)/NiFe (4 nm) bilayers, where the IrMn thickness ($t_{IrMn}$) ranges from 5 to 25 nm, demonstrating

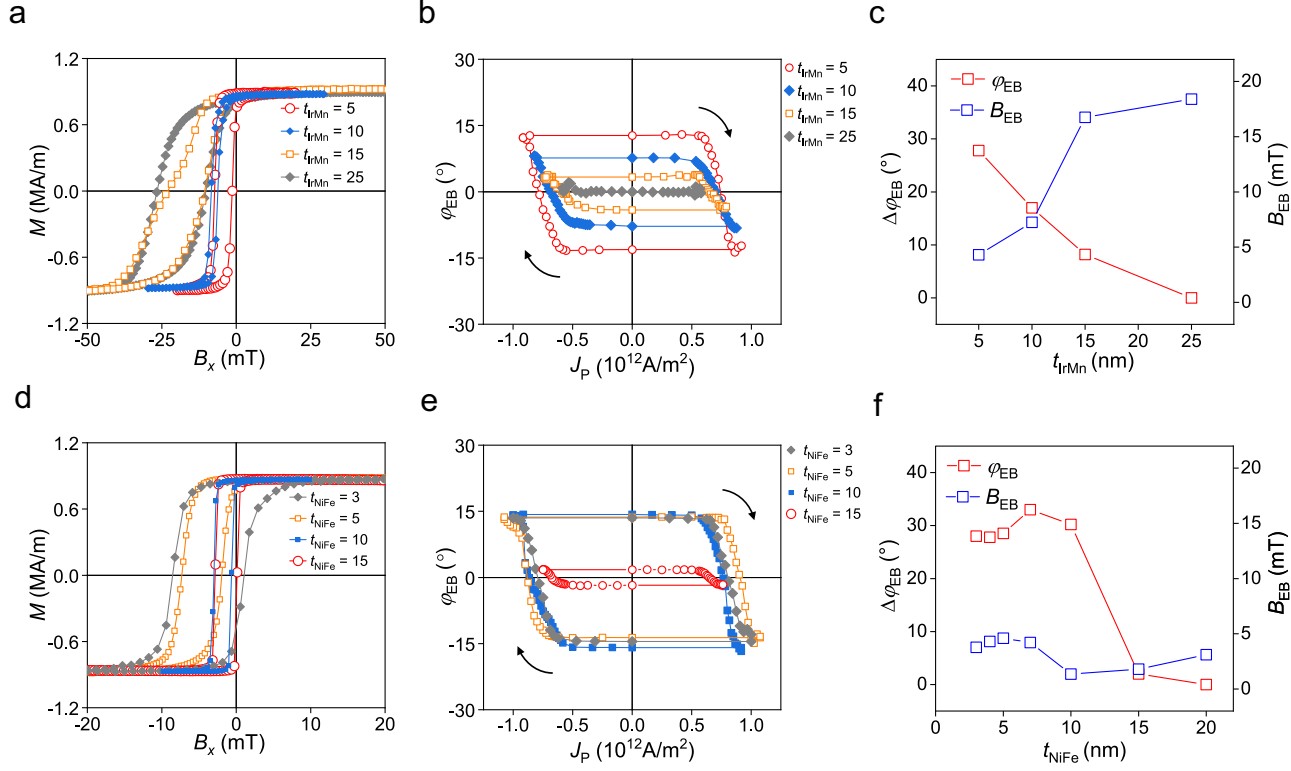

**Fig. 4 Thickness dependence of SOT-induced exchange bias switching. a–c.** Hysteresis loop measured using a magnetic field along the x-axis, $B_x$ (**a**), the $\varphi_{EB}$ versus current density ($J_P$) curves, where the arrows denote the sweeping direction of $J_P$ (**b**), $\Delta\varphi_{EB}$ [$= \varphi_{EB}$ $(-I) - \varphi_{EB}$ $(+I)$] and $B_{EB}$ as a function of IrMn thickness $t_{IrMn}$ (**c**) of the IrMn ($t_{IrMn}$)/NiFe (4 nm) samples with different $t_{IrMn}$'s ranging from 5 to 25 nm. **d–f.** Hysteresis loop measured using $B_x$ (**d**), the $\varphi_{EB}$ versus $J_P$ curves, where the arrows denote the sweeping direction of $J_P$ (**e**), the $\Delta\varphi_{EB}$ and $B_{EB}$ versus $t_{NiFe}$ (**f**) of the IrMn (5 nm)/NiFe ($t_{NiFe}$) samples with different $t_{NiFe}$'s ranging from 3 to 20 nm.

that the exchange bias field ($B_{EB}$) increases with an increase in $t_{IrMn}$. This is attributed to the enhancement of the AFM anisotropy for a thicker $t_{IrMn}$[48–50]. Figure 4b plots the $\varphi_{EB}$ versus current density $J_P$ curves for samples with different $t_{IrMn}$; the maximum value of $\varphi_{EB}$ achieved by SOT is gradually reduced as $t_{IrMn}$ is increased. Figure 4c summarizes the results showing that the maximum variation of the $\varphi_{EB}$ value [$\Delta\varphi_{EB} = \varphi_{EB}$ $(-I) - \varphi_{EB}$ $(+I)$] decreases with increasing the magnitude of $B_{EB}$. This indicates that the SOT-induced rotation of the exchange bias is impeded by the AFM anisotropy developed in the x-direction. Therefore, $\varphi_{EB}$ can be further enhanced by reducing the AFM anisotropy as long as the exchange-coupling with the FM is preserved.

We also examine the dependence of $B_{EB}$ and $\varphi_{EB}$ on the NiFe thickness ($t_{NiFe}$) in IrMn (5 nm)/NiFe ($t_{NiFe}$) bilayers, where $t_{NiFe}$ is varied from 3 to 20 nm. Figure 4d, e shows the hysteresis loops and the $\varphi_{EB}$ versus $J_P$ curves of the samples, respectively (see Supplementary Note 11 for the full set of data). As summarized in Fig. 4f, $B_{EB}$ remains unchanged for $t_{NiFe}$'s up to 7 nm and decreases slightly when $t_{NiFe}$ exceeds 10 nm, which is the general trend of the reduction of $B_{EB}$ for thicker FMs[48,51]. Interestingly, $\Delta\varphi_{EB}$ remains constant around 30° when $t_{NiFe}$ ranges from 3 to 10 nm, and it significantly reduces for a $t_{NiFe}$ larger than 15 nm. This is inconsistent with the aforementioned assumption that spin currents generated in IrMn exert SOTs on NiFe, in which case the magnitude of SOT (or the maximum $\varphi_{EB}$ value) should decrease with an increase in $t_{NiFe}$ because spin currents are mostly absorbed at the FM interface when injected into an FM layer. This result suggests that the SOT-induced AFM switching is an interface phenomenon; therefore, we infer that the spin current gives spin torques to the interfacial uncompensated

AFM moments, rotating the magnetization directions of the exchange-coupled IrMn and NiFe simultaneously.

**Memristive behavior of exchange bias switching.** We finally discuss memristive characteristics based on SOT-induced exchange bias switching in a reversible and non-volatile manner. Figure 5a shows minor $\varphi_{EB}$–$J_P$ curves of the IrMn (5 nm)/NiFe (4 nm)/Ta (1.5 nm) structure with a 4-μm-wide Hall bar. As shown in the measurement sequence illustrated in Fig. 5b, we first apply an initializing current pulse $J_{P,ini}$, of $-7.4 \times 10^{11}$ A/m² to set $\varphi_{EB} = +22°$ and then measure $R_H$ while sweeping $J_P$ between $-7.4 \times 10^{11}$ A/m² and the positive maximum $J_P$ [$J_{P(+max)}$]. The measurement is repeated as we increase $J_{P(+max)}$ from $5.2 \times 10^{11}$ to $7.4 \times 10^{11}$ A/m². This result demonstrates that multiple $\varphi_{EB}$ values between ±22° can be obtained according to the magnitude of $J_{P(+max)}$. Similar results of minor loops are observed when sweeping $J_P$ between $+7.4 \times 10^{11}$ A/m² and the negative maximum $J_P$ (Supplementary Fig. S12). We test whether the memristive feature is maintained in nanoscale devices. Figure 5c shows the minor $\varphi_{EB}$–$J_P$ curves of the IrMn (5 nm)/NiFe (5 nm) structure with a 500-nm-wide Hall bar, as measured with experimental procedures similar to that shown in Fig. 5a, b. Multi-level $\varphi_{EB}$ values are successfully achieved in the 500 nm device, comparable to those in the 4 μm sample. This implies that the gradual change of $\varphi_{EB}$ in IrMn/NiFe bilayers is due to the collective rotation of the AFM moments and the exchange-coupled FM moment, which is distinct from the previous results based on AFM domain wall motions[28,29,39,52]. The scalable memristive characteristics can facilitate electrically controlled multi-level spintronic devices for neuromorphic computing.

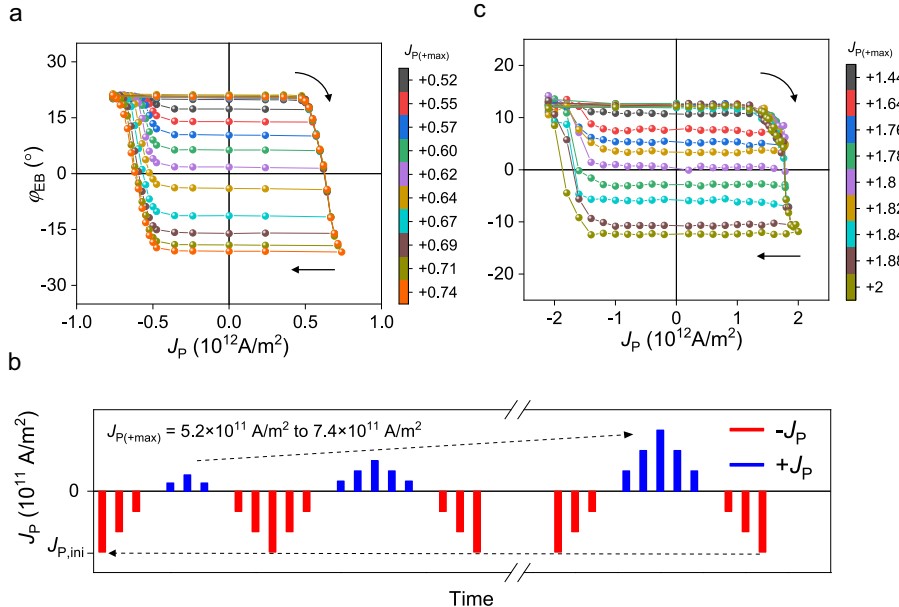

**Fig. 5 Memristive behavior of exchange bias switching. a** Minor $\varphi_{EB}$ versus $J_P$ curves for the IrMn (5 nm)/NiFe (4 nm)/Ta (1.5 nm) sample with a 4-μm-wide Hall bar. The arrows denote the sweeping direction of $J_P$, **b** Schematics of the measurement sequence; $J_{P,ini}$ is the initializing current pulse of $-7.4 \times 10^{11}$ A/m$^2$, and $J_{P(+max)}$ is the positive maximum $J_P$. Minor loops are consecutively measured while sweeping $J_P$ between $-7.4 \times 10^{11}$ A/m$^2$ and $J_{P(+max)}$, which increases from $5.2 \times 10^{11}$ to $7.4 \times 10^{11}$ A/m$^2$ mA. **c** Minor $\varphi_{EB}$ versus $J_P$ curves for the IrMn (5 nm)/NiFe (5 nm) sample with a 500-nm-wide Hall bar. The arrows denote the sweeping direction of $J_P$.

## Discussion

We have demonstrated reversible current-induced manipulation of the exchange bias in IrMn/NiFe bilayer structures. We observe that the SOT caused by the spin Hall effect in IrMn effectively controls the exchange bias up to ±22°. The maximum rotation angle of the exchange bias achieved by SOT is independent of the NiFe thicknesses up to 10 nm, indicating a critical role of interfacial uncompensated AFM moments that mediates the spin torque to the entire AFM and exchange-coupled FM moments. Moreover, memristive behavior, the gradual manipulation of the exchange bias according to the polarity and amplitude of the electric current, can be observed in a 500 nm device. Our results demonstrating the electrical control of the AFM moment in a reversible and non-volatile manner paves the way for the realization of nanoscale AFM-based spintronics for neuromorphic applications.

## Methods

**Sample preparation**. Samples of IrMn/NiFe, IrMn/NiFe/Ta, NiFe/IrMn, and Ta/NiFe structures were deposited on high-resistivity Si substrates using ultrahigh-vacuum magnetron sputtering with a base pressure of less than $4.0 \times 10^{-6}$ Pa. The compositions of the deposited layers are $Ir_{25}Mn_{75}$ and $Ni_{76}Fe_{24}$, which are analyzed by inductively coupled plasma optical emission spectroscopy (Supplementary Note 1). The resistivity of each layer is 316 μΩ cm for IrMn and 51.3 μΩ cm for NiFe layer (Supplementary Note 9). During the sputtering process at room temperature, a magnetic field of 15 mT was applied to induce uniaxial anisotropy of the NiFe layer. A capping layer of MgO (3.2 nm)/Ta (2 nm) was used to protect samples from oxidation. The metal layers were grown by DC sputtering (30 W) with a working pressure of 0.4 Pa, while the MgO layer was deposited by RF sputtering (150 W) at 1.33 Pa. After the deposition step, the samples were annealed at 200 °C for 40 min in a vacuum with a magnetic field of 100 mT to develop the exchange bias. The magnetic hysteresis loop was measured using a vibrating sample magnetometer. For the electrical measurements, a Hall bar structure with a 4 μm (or 500 nm) width was patterned by photo- (or e-beam-) lithography and Ar ion etching. Electrical contacts were formed by the deposition of Ru (50 nm) and a subsequent lift-off process.

**Electrical measurements**. Planar Hall resistance ($R_H$) was measured with a reading current of 100 μA while rotating the samples in the $x$–$y$ plane under an external magnetic field of 100 mT. For the current-induced AFM switching measurements, $R_H$ was measured with a reading current of 100 μA after 500 ms of each current pulse ($J_P$) application of 30 μs width. The constant offset of $R_H$, which may be caused by the misalignment of the Hall cross, is removed such that $R_H = 0$ corresponds to the magnetization aligned to the $x$-direction ($\varphi_m = 0°$). All switching measurements were performed at room temperature without an external magnetic field.

## Data availability

The data that support the findings of this study are available from the corresponding author upon reasonable request.

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

## Acknowledgements
This work was supported by the National Research Foundation of Korea (NRF-2015 M3D1A1070465 and 2020R1A2C2010309). K.-J.K. and B.-G.P. acknowledge support from KAIST-funded Global Singularity Research Program for 2021.

## Author contributions
B.-G.P. planned and supervised the study. J.K., J.-G.C., and J.R. fabricated devices. J.K. and J.R. performed electrical and magnetic measurements with help from S.L., J.P., T.L., and K.-J.K. H.J. and Y.S.J. performed the thermal conductivity measurement. All authors discussed the results, and J.K., J.R., and B.-G.P. wrote the paper.

## Competing interests
The authors declare no competing interests.
