## [Peer Review File · Nature Communications]

Reviewers' Comments:

Reviewer #1:

Remarks to the Author:

Kang et al. report current induced resistance changes of IrMn/NiFe bilayers, which are associated with a SOT induced modification of the easy axis of the AFM layer.

These are very interesting experimental results, which in principle are suitable for publication in Nat. Commun. However, some clarifications concerning the discussion of the relevant physical effects are required.

The term "easy axis" could be misleading in this manuscript. "Easy axis" should be reserved for the preferential orientation of the magnetization/ Néel vector due to magnetocrystalline anisotropy or geometrical effects. Instead, what is considered in the manuscript is presumably exchange bias.

The experimental observations are explained by a SOT induced rotation of the NiFe moment, which is followed by the rotation of the exchange coupled IrMn moment. However, it is also required to discuss in more detail why this happens only, if initiated by current pulses and not upon magnetic field driven rotation of the IrMn magnetization. Indeed, the authors mention that the sample temperature increases due to the current injection and propose that this may reduce the AFM anisotropy. However, in this framework it is important to estimate the temperature of the Hall bar during the current pulse. The used current density of about 1×10^8 A/cm² is huge and will result in significant heating. With an oscilloscope, it should be possible to measure the sample resistance during the pulse. As the sample is metallic, its resistance will increase and by comparing with a R(T) measurement of the sample the temperature during the pulse can be estimated quite precisely. This is important, as the blocking temperature of thin IrMn is low (70 C as measured here: J. Appl. Phys. 107, 09D728 (2010)). I expect that during the current pulse the sample temperature is much higher. Thus, it would be possible to explain the observed resistance modification by setting a new exchange bias with each current pulse. This could as well explain the observed saturation of the observed effects above certain current densities, as the "field cooling" with decreasing current amplitude would dominate the applied exchange bias and would always happen at the blocking temperature.

Furthermore, there are some less important issues, which require clarification:

On line 85, the authors should mention, that the thin films are polycrystalline.

In line 105, the current density should be given. It is also an important information at which current density the structures usually break.

What is the basis of assuming that R_H is dominated by NiFe (line 114)? Is the AMR of IrMn known?

Why does the polycrystalline single NiFe layer have an easy axis as mentioned in line 128? I assume this is geometrically induced parallel to the Hall bar. Please mention this explicitly.

On line 136, a potential similarity to Néel SOT is discussed. This is misleading, as a NSOT requires a special crystallographic symmetry, which only CuMnAs and Mn₂Au show. Thus this mechanism is not possible for IrMn.

On lines 182-185, the role of uncompensated moments at the interface is discussed. In this framework, references to older work on exchange bias such as Phys. Rev. B 35, 3679 would be helpful.

Reviewer #2:

Remarks to the Author:

The manuscript by Kang et al. reports electrical reversible manipulation of the exchange bias direction in IrMn/NiFe structure. The authors observe a change in the exchange bias direction of

IrMn/NiFe by applying an electric current into the structure and the change is maintained after the current is returned to zero. From a systematic investigation using several structures with reversed stacking order, Ta cap, various IrMn thicknesses, and NiFe layer thicknesses, they attribute their observation to the spin current generated in the IrMn layer through the spin Hall effect and exerted on the interfacial uncompensated moment of IrMn. In addition, they demonstrate a controllability of the change in an analog (memristive) fashion with good scalability and conclude that the established scheme is promising for neuromorphic computing.

The exploration of new functionalities of antiferromagnetic materials is an active field in the spintronics, recognized as the antiferromagnetic spintronics, and I think this study adds a new option to this field as it provides a new means to take advantage of the natures of antiferromagnets. Furthermore, as the authors claim, I think this work may be important from technological point of view, because the neuromorphic computing requires scalable and highly controllable memristive devices. In these regards, I think the results presented in this manuscript "potentially" deserves to be published in high impact journals such like the Nature Communications. Nevertheless, in my opinion, the current manuscript has several critical issues which need to be addressed satisfactorily for further consideration. The details are listed below:

[Major issues]

1. Terminology

The authors describe that the easy axis is electrically manipulated, but this sounds to me odd. The easy axis represents the energetically favorable direction of magnetic moments that is determined by the local atomic structures or the global shape of the magnet; for example, the easy axis of Co is along the c axis and easy axis of bar magnet is along the longitudinal direction. Meanwhile, I believe the phenomena observed in this work does not accompany any change in atomic structures and, of course, the shape of the device. Instead, in my understanding, what they observe is just a change in the exchange bias direction. Therefore, I think the authors should reconsider the meaning of "easy axis" and revise the manuscript accordingly.

2. Quantification of several effects

If the authors wish to present these results in high impact journal like Nature Communications, I think more quantitative investigations are mandatory. For example, the temperature rise due to the Joule heating and its comparison to the blocking temperature are essential because the observed phenomena should be strongly dependent on the temperature. I also think that quantification of the current distribution in each layer, Oersted field, spin-orbit torque (sign and magnitude of the spin Hall angle) of IrMn and Ta is necessary; otherwise, their interpretation cannot be supported well. The value of current density in addition to the value of current should be also presented because the current density is more relevant when one considers the underlying physics.

3. Physical scenario

I agree that the inference of the authors that "the obtained result originate from a spin current generated in IrMn and exerted on uncompensated moment of IrMn" is one of the possible scenarios, but I am wondering if this is the only one. I would suggest that the authors should consider other effects such as the spin anomalous Hall effect of NiFe and interfacial phenomena including the Rashba-Edelstein effect. Also, I think the spin torque acting on the magnetic moment of NiFe near the interface as well as on the uncompensated moment of IrMn should also play a certain role and cannot be excluded.

4. Comparison to previous study

I think the significance of this manuscript will be even improved by adding comparative discussion to previous related studies on antiferromagnet-ferromagnet structures after presenting their results. In particular, this work is similar to Ref. 26 (Lin et al., Nature Mater. 2019) where the electrical manipulation of exchange bias by SOT is reported although the direction of the exchange bias is different from the present manuscript. As far as I understand, explicit explanation is not given in Ref. 26 and I am wondering if the authors can provide some speculation to the underlying mechanism for the results reported in Ref. 26.

5. Information of the studied material system

Information of the studied material system is not sufficiently presented. At least, the following

information should be presented so that the readers can perform replication study: composition of IrMn and NiFe, temperature of the stage during the deposition, underlayer (if any). In addition, the crystallographic structure, crystal orientation, and nanoscale texture of IrMn are also essential information because these factors are known to strongly relate to the exchange bias. Description on the possible magnetic structure of IrMn should be also added because it is known to crucially affect the spin transport properties of IrMn. In fact, it is reported that IrMn₃ generates out-of-plane component of spin accumulation [<https://doi.org/10.1103/PhysRevApplied.12.064046>], whereas such effect is not discussed in this manuscript.

[Minor issues]

6. "memu" is used as a unit of magnetization curve but this is not suitable because, firstly, emu is not a unit of magnetization, secondly, the quantity with the unit of emu depends on the sample size whereas the information of the sample size is not described, and thirdly this manuscript adopts SI or MKSA unit system whereas "emu" is a unit of cgs system.

7. "inversely proportional to" in line 167 is not appropriate because the result seems not show the relation of $y = 1/x$. This part should be rephrased.

8. "coherently" is used several times without any definition and I could not understand its meaning after going through the manuscript. However, I think the meaning should be clearly defined somewhere.

Reviewer #3:

Remarks to the Author:

This work reports a repeatable change of the exchange bias direction driven by electric current. This change seems to be reversible and it leads to a memristive behaviour that is also present in sub-micron wide stripes. The authors claim that all is due to spin orbit torques producing a rotation of the AFM easy axis (and/or amplitude), which makes the finding potentially quite interesting. On the other hand, the origin of the effect is unclear to me and the authors need to provide more measurements to build a stronger case and a stronger discussion.

A.- For instance, could the effect be the result of a combination of temperature plus Oersted field from the current achieving some sort of local annealing? The current densities are quite large, even larger than 10^{12} A/m² in the 500nm device. What happens for instance in Figure 1d if they go beyond 30 mA? The effect does not seem saturated. If they cannot go beyond 30 mA, because they destroy the sample, maybe this is an indication of a very relevant Joule heating. The authors acknowledge there may be some heating in line 149. In figure 2b they show that the effect grows when they have a 1.5nm Ta layer on top, claiming that the Ta layer diminishes the Oe field. I presume they say that because there is less current density. But, assuming the Ta is not oxidized (they mention a MgO/Ta capping layers, that I guess it is deposited in the same sputtering run), more current will flow through the Ta and NiFe layer than on the IrMn and the Oe field in the NiFe will be more intense. Fig. 2b is an interesting result but does not completely rule out the possibility of the whole effect being simply current induced annealing.

In order to rule out this possibility, it is not enough with a simulation of a quick calculation following a theoretical model. The real sample may have a large interface thermal resistance with the substrate and the temperature could be larger than what any simulation would show. As the authors are using pulses in the microsecond range (which in terms of heating is pretty much like a DC current) and they do not need transmission lines to apply the pulses, perhaps the best solution is to repeat (at least) one of the measurements on a cryostat, even if it is only at 77 K. If the effect is still there, it would give confidence that it is not a current induced annealing of the sample.

B.- My other big concern is that they are inferring a change in the exchange bias direction in the whole length of the strip, by measuring the PHE, which is only sensitive to what is happening in the Hall cross. Could they be measuring a local change that is happening only at the cross and not in the rest of the stripe. The authors should supply also AMR measurements (along the length of the strip), perhaps even using the existing longitudinal contacts if they do not want to process new Hall bars. It would be useful to contrast with a AMR hysteresis loop that the angle they are measuring with PHE is actually a change of the exchange bias direction along the entire nanostrip.

C.- Being the IrMn such a popular material, it is incredible that nobody has seen this effect before.

And there are plenty experiments with similar current densities in spin valves with IrMn. Do the authors find the same behaviour using other ferromagnetic material on top of the IrMn, such as FeCo for instance?

D.-The authors suggest that a spin current arising from the IrMn layer exerts SOTs on uncompensated antiferromagnetic moments at the interface and then rotate the antiferromagnetic moments" but where is the evidence that, if there is a spin current, it is acting on the uncompensated moments? At the most, this is a hypothesis.

In general, the authors seem to perceive the anisotropy of the IrMn and the Exchange Bias field as the same thing. For instance, in lines 66-68, the authors state "Fi_{AFM} diminishes with an increase in the IrMn thickness, indicating that the SOT-induced rotation of the AFM easy axis is hindered by the AFM anisotropy"- I don't understand this phrase - and then continues "which increases with its thickness". How do they know that the anisotropy of the IrMn increases with thickness in their samples? In figure 3a, they see an increase of the EB field with thickness of the IrMn, but I do not see why that may be a direct indication of the strength of the anisotropy. Thicker films should lead to a larger volume of the activated grains in the IrMn, but I do not see the straight link with the anisotropy.

In fact, throughout the article, the claim is the change of the anisotropy axis in the IrMn with the electric current, but the observation is a change in the direction of the EB with the electric current and I am not sure it is the same thing. They could be triggering a rotation of pinned rotatable uncompensated moments only at the interface (as they say in the abstract) or annealing with current some of the IrMn grains. Therefore, in my view, throughout the paper, they should speak about current induced persistent rotation of the Exchange Bias direction.

E.-When they mention a current value (i.e 15 mA) they should give also the current density value, at least the first time they mention that value of current. In general, throughout the paper is better to give also values of the current density.

F.-In the experiment described in lines 145 to 147, where they grow the NiFe/IrMn, is not that the same as changing the direction of the current in the IrMn/NiFe sample? I do not quite see why it is different and why it is an experiment that proves anything.

G.-In Figure 3b, can they measure to slightly higher current densities? It would also be nice (like in the rest of the figures) to show few points of the saturation of the effect and, in this case, plot all the curves with the same maximum and minimum current densities. My guess here is that the authors cannot plot all the curves with the same maximum current density because of the heating. For the same current density, thicker samples heat up more, and my guess is that they may destroy the samples. Is this the case?

H.-Figure 3e is quite puzzling for me. The fact that the effect does not depend at all with the thickness of the NiFe does not quite fit with current induced thermal annealing, but I don't quite understand it either from the point of view of a SOT. For a fixed exchange bias field, the thicker the NiFe layer, the larger the energy involved. So, how can the same sequence of current pulses deliver the same Exchange Bias rotation regardless of the thickness of the NiFe? If the authors understand it, they should elaborate a bit more their explanation. In line 183, they conclude that it is an interface phenomenon but then, it would decrease inversely proportional to the thickness of NiFe. In any case, I suggest to include two measurements:

H(a).- The results for NiFe thickness 15 and 20 nm. I guess at some point the thickness of the NiFe must be detrimental.

H(b).- This figure is one of the reasons I am suggesting, on top of the PHE measurements that the article provide, AMR measurements to see the hysteresis loop of the entire length of the strip, and rule out that what the authors are seeing is just a local fluctuation of the magnetization in the cross.

Further corrections:

- 1.-Title: As the claimed effect is achieved with electric current, perhaps the title should be "Current induced manipulation of..."
- 2.-Line 16: "To date, the manipulation of antiferromagnetic moments..." perhaps should clarify that is manipulation by electric means or by electric current.
- 3.-Line 19: "...electrical manipulation..." by "current induced manipulation"
- 4.-Paragraph from line 51 to 58 is very difficult to understand.
- 5.-Line 63. "...and the associated exchange bias between up to ± 22 degrees". What does this mean?

- 6.-Line 85. "... by deposition" or "by sputtering deposition"?
- 7.-Figure 1d. As in figure 1c, put the field B at which this measurement is taken. I guess $B=0$ mT.
- 8.-Figure 1d. I think it is better to give the values of current in the x-axis as a current density.
- 9.-Line 110: "...demonstrating the electrical modulation of ϕ_m of 15° in a reversible manner". I don't think 'modulation' is the right word. Consider something in the lines of 'demonstrating a repeatable commutation of $F_{i,m}$ between $\pm 15^\circ$ with electric current'.
- 10.-Line 121: "The spontaneous recovery...", they mean the recovery when the field is reduced to zero, so it is not really spontaneous. Remove 'spontaneous'?
- 11.-Line 161: In reference to Figure 3 it mentions that the thickness of IrMn goes from 5 to 25nm, but in the figure they only show 5, 10 and 15 nm. One has to go to supplementary material to see the 20 and 25 nm. I believe in delivering most of the important information in the main text, rather than in the Supplementary Information. Therefore, it would be nice if they could somehow include the 20 and 25 nm IrMn thickness in the figure 3b.
- 12.-Figure 4: Plot the x-axis in terms of current density.
- 13.-Line 201: "This implies...". I do not see why the experiments of the memristive behaviour imply collective rotation of the AFM easy axis. Perhaps after all the corrections are addressed, this point may become clearer.
- 14.-Although I am not a native speaker myself, I think the manuscript requires a general English revision.

Dear Reviewers,

We appreciate your comments and valuable queries, which have helped us improve the clarity and quality of our manuscript. Given below are detailed point-by-point responses to your questions and suggestions. The corresponding modifications are incorporated in the revised manuscript (marked in blue). We believe our manuscript has been improved significantly and now deserves publication in *Nature Communications*.

Yours sincerely,

Byong-Guk Park on behalf of all co-authors

[Reviewer #1]

Kang et al. report current induced resistance changes of IrMn/NiFe bilayers, which are associated with a SOT induced modification of the easy axis of the AFM layer. These are very interesting experimental results, which in principle are suitable for publication in Nat. Commun. However, some clarifications concerning the discussion of the relevant physical effects are required.

Response) We appreciate the reviewer's comment of "These are very interesting experimental results, which in principle are suitable for publication in Nat. Commun.". However, she/he points out that "some clarifications concerning the discussion of the relevant physical effects are required". We respond to the reviewer's comments with additional experiments described below, which will hopefully alleviate the reviewer's concerns.

The term "easy axis" could be misleading in this manuscript. "Easy axis" should be reserved for the preferential orientation of the magnetization/Néel vector due to magnetocrystalline anisotropy or geometrical effects. Instead, what is considered in the manuscript is presumably exchange bias.

Response) We acknowledge the reviewer's comment that "The term easy axis could be misleading." So, we reworded the term "easy axis" to "exchange bias" in the revised manuscript.

The experimental observations are explained by a SOT induced rotation of the NiFe moment, which is followed by the rotation of the exchange coupled IrMn moment. However, it is also required to discuss in more detail why this happens only, if initiated by current pulses and not upon magnetic field driven rotation of the IrMn magnetization. Indeed, the authors mention that the sample temperature increases due to the current injection and propose that this may reduce the AFM anisotropy. However, in this framework it is important to estimate the temperature of the Hall bar during the current pulse. The used current density of about 1×10^8 A/cm² is huge and will result in significant heating. With an oscilloscope, it should be possible to measure the sample resistance during the pulse. As the sample is metallic, its resistance will increase and by comparing with a $R(T)$ measurement of the sample the temperature during the pulse can be estimated quite precisely. This is important, as the blocking temperature of thin IrMn is low (70 C as measured here: J. Appl. Phys. 107, 09D728 (2010)). I expect that during

the current pulse the sample temperature is much higher. Thus, it would be possible to explain the observed resistance modification by setting a new exchange bias with each current pulse. This could as well explain the observed saturation of the observed effects above certain current densities, as the “field cooling” with decreasing current amplitude would dominate the applied exchange bias and would always happen at the blocking temperature.

Response) We appreciate the reviewer for this critical comment. To rule out thermal effects as a possible cause of the results, we performed additional experiments: (i) measurement of the blocking temperature of an IrMn/NiFe structure, (ii) estimation of the temperature rise during the switching experiments, (iii) current-induced switching experiment at a low temperature, (iv) calculation of Oersted field by the injecting current. As demonstrated below, the experimental results corroborate that the current-induced switching of the exchange bias is not due to thermal effects.

(i) Blocking temperature of an IrMn/NiFe structure

We first measure the blocking temperature of the IrMn (5 nm)/NiFe (4 nm) sample that is used for the measurements shown in Fig. 1 of the main text. Figures R1a,b shows magnetization curves measured at temperatures ranging from 100 K to 400 K. The temperature dependence of the exchange bias field (B_{EB}) that is determined by the hysteresis shift is plotted in Figure R1c, demonstrating that the B_{EB} appears up to 380 K and disappears at temperatures greater than 385 K. The blocking temperature at which the exchange bias vanishes is 385 K for the IrMn (5 nm)/NiFe (4 nm) sample.

Figure R1 | Temperature dependence of exchange bias in the IrMn (5 nm)/NiFe (4 nm) sample. a-b, Magnetization loops at different temperatures of 100~340 K (a) and 360~400 K (b). **c,** Exchange bias field B_{EB} versus temperature. B_{EB} is determined by the hysteresis shift in Figure R1a-b.

(ii) Temperature rise during the switching experiments

We next estimate the sample temperature during the current pulse application by measuring the longitudinal resistance R_{xx} of the IrMn (5 nm)/NiFe (4 nm) sample using an oscilloscope. Figure R2a shows the time-dependent R_{xx} while applying current pulses I_P of different magnitudes ranging from 0.1 mA to 27 mA (equivalent to $8.4 \times 10^{11} \text{ A/m}^2$). Here, the pulse width is 30 μs , identical to that used in the switching experiments of the original manuscript. As shown in the inset, the R_{xx} increases with the current magnitude. Note that the different pulse shapes are due to the different pulse modes according to the current range of the current source used in our experiment (Keithley 6221). Figure R2b shows the change in R_{xx} , $\Delta R_{xx} [= R_{xx}(I_P) - R_{xx}(I_P = 0.1 \text{ mA})]$ as a function of the I_P^2 , indicating the ΔR_{xx} is primarily due to Joule heating by the pulse current. Then, we measure the temperature dependence of R_{xx} and plot ΔR_{xx} versus temperature in Fig. R2c. By comparing those two graphs (Figs. R2b,c), we estimate the sample temperature when applying a current pulse density J_P as shown in Figure R2d. This indicates that the sample temperature increases to $354 \pm 3.5 \text{ K}$ when the critical switching current $(7.4 \pm 0.28) \times 10^{11} \text{ A/m}^2$ is applied (Fig. 3b of the original manuscript). This is lower than the blocking temperature of the sample (385 K), indicating that the current-induced manipulation of the exchange bias in our experiments is not due to Joule heating.

We included the following sentences regarding the results of thermal effect on page 8 of the revised manuscript and Supplementary Note 6,

“We note that the sample temperature increases due to the current injection, but not above the blocking temperature (Supplementary Note 6), which may reduce the AFM anisotropy and thereby assists the rotation of the AFM moment.”

Figure R2 | Joule heating effects in the IrMn (5 nm)/NiFe (4 nm) sample. **a**, Time (t)-dependent resistance measurements using oscilloscope. Here, a current pulse I_P ranges from 0.1 mA to 27 mA and the pulse width is 30 μ s. Inset shows the magnification of the time-dependent resistance data. **b**, Change in longitudinal resistance R_{xx} (ΔR_{xx}) as a function of I_P^2 . The R_{xx} value of each temperature is obtained by averaging the data from $t = 20 \mu$ s to $t = 30 \mu$ s. **c**, Temperature-dependent ΔR_{xx} of the sample. **d**, Estimated sample temperature T as a function of current density J_P . The line represents a fitting curve of $T=300+ 98.4J_P^2$

(iii) Current-induced switching experiment at a low temperature

To further elucidate Joule heating effect, we examine the current-induced manipulation of the exchange bias direction of the IrMn (5 nm)/NiFe (4 nm) sample at low temperatures. We performed current-induced exchange bias switching experiment at temperatures between 100 K and 300 K using the same measurement protocols described in the main text. Figure R3a shows the switching results of R_H versus pulse current J_P , demonstrating that the switching behaviour for all measuring temperatures is similar to that measured at 300 K (Fig. 3b of the original manuscript). By comparing the maximum R_H value of each temperature with planar Hall resistance measured under an external magnetic field of 200 mT (Fig. R3b), we extract the rotation angle of the exchange bias (ϕ_{EB}) for each temperature (Fig. R3c). This demonstrates

that the current-induced modulation of φ_{EB} is obtained regardless of the measuring temperature down to 100 K, which is much lower than the blocking temperature. This result confirms that the Joule heating effect is not a possible origin of the current-induced manipulation of exchange bias.

We included the following sentence regarding the results of switching experiment at a low temperature on page 8 of the revised manuscript and Supplementary Note 7.

“Similar switching behaviour is obtained at low temperatures down to 100 K, much lower than the blocking temperature, confirming that Joule heating effect is not a possible origin of the current-induced manipulation of exchange bias (Supplementary Note 7).”

Figure R3 | Current-induced manipulation of the exchange bias at low temperatures. a, The Hall resistance (R_{H}) of the IrMn (5 nm)/NiFe (4 nm) sample measured after applying an in-plane current pulse J_{P} at various temperatures. **b,** Planar Hall resistance R_{H} versus azimuthal angle of a magnetic field (φ_{B}) of 200 mT at different temperatures. **c,** The rotating angle of the exchange bias φ_{EB} versus J_{P} curves at various temperatures. The arrows denote the sweeping direction of J_{P} .

(iv) Oersted field by the injecting current

We also check if Oersted field caused by injecting current affects the current-induced modulation of the exchange bias. In this regard, we first estimate the current distribution of each layer by measuring the resistance R_{xx} of the devices with different IrMn and NiFe thicknesses. Here, NiFe thickness t_{NiFe} ranges from 3 to 15 nm in IrMn (5 nm)/NiFe (t_{NiFe}) structures and IrMn thickness t_{IrMn} ranges from 5 to 25 nm in IrMn (t_{IrMn})/NiFe structures. Figure R4a shows the $(1/R_{\text{xx}}) \cdot L/W$ values of the samples as a function of t_{NiFe} and t_{IrMn} , where L and W is the length and width of the Hall-bar device, respectively. From the results, we extract the resistivity of each layer; 316 $\mu\Omega\text{cm}$ for IrMn and 51.3 $\mu\Omega\text{cm}$ for NiFe layer. This indicates

that in the NiFe (4 nm)/IrMn (5 nm) bilayers, 83.1% and 16.9% of the current flows through the NiFe and IrMn layers, respectively. Moreover, in the IrMn (5 nm)/NiFe (4 nm)/Ta (1.5 nm) sample, where the resistivity of Ta is 353.8 $\mu\Omega\text{cm}$, 16.2%, 79.5%, 4.3% of the current flows through the IrMn, NiFe, and Ta (1.5 nm) layers, respectively.

Taking the current distributions into account, we calculated an Oersted field acting on NiFe layer in the IrMn (5 nm)/NiFe (4 nm) and IrMn (5 nm)/NiFe (4 nm)/Ta (1.5 nm) samples [Nat. Nanotech. 12, 980-986 (2017)]. Figure R4b,c shows the Oersted fields along the y and z direction ($B^{\text{Oe},y}$ and $B^{\text{Oe},z}$) generated in the middle of the NiFe layer of the samples. Here, we use a current density of $7.4 \times 10^{11} \text{ A/m}^2$, which is the switching current density of the sample. It is found that $B^{\text{Oe},y}$ is ~ 0.68 mT in the IrMn (5 nm)/NiFe (4 nm) sample and ~ 0.55 mT in the IrMn/NiFe/Ta sample. This allows us to exclude the Oersted effect from the main cause of the current-induced modulation of the exchange bias for the following two reasons; first, a larger rotation angle of exchange bias φ_{EB} is observed in the IrMn/NiFe/Ta sample, where the current-induced Oersted field is smaller. Second, such a small Oersted field of less than 1 mT cannot make any noticeable change in the magnetization direction of the exchanged coupled IrMn/NiFe sample as demonstrated in Fig. R4d, where planar Hall resistance R_{H} measured while a rotating magnetic field of 1 mT remains almost constant as compared to that of 200 mT. Note that the *cos* component of R_{H} for 1 mT is probably due to the polar angle induced by a slight misalignment of the sample.

We included the above results as Supplementary Note 8 in the revised manuscript.

Figure R4 | Current distribution and Oersted field. a, $(1/R_{xx}) \cdot (L/W)$ as a function of NiFe thickness t_{NiFe} or IrMn thickness t_{IrMn} in IrMn/NiFe bilayer structures. **b,c** Calculated Oersted fields along the y and z direction ($B^{\text{Oe},y}$ and $B^{\text{Oe},z}$) in the middle of NiFe layer of the IrMn (5 nm)/NiFe (4 nm) sample (**b**) and the IrMn (5 nm)/NiFe (4 nm)/Ta (1.5 nm) sample (**c**). Here, a current density is $7.4 \times 10^{11} \text{ A/m}^2$. **d**, Planar Hall resistance R_H of the IrMn/NiFe sample measured with external magnetic fields of 200 mT and 1 mT.

1. On line 85, the authors should mention, that the thin films are polycrystalline.

Response) We revised the sentence; “We fabricate a (111) textured polycrystalline IrMn (5 nm)/NiFe (4 nm) bilayer by sputtering deposition” on page 4 in the revised manuscript.

2. In line 105, the current density should be given. It is also an important information at which current density the structures usually break.

Response) As suggested by the reviewer, we revised “current” to “current density” throughout the manuscript, and included the sentence about breakdown of our device “Note that the device tends to break down when the current density exceeds $1.0 \times 10^{12} \text{ A/m}^2$.” on page 6.

3. What is the basis of assuming that R_H is dominated by NiFe (line 114)? Is the AMR of IrMn known?

Response) To clarify if the R_H of the IrMn/NiFe sample is mainly dominated by the NiFe layer, we measure the planar Hall effect (PHE) of two samples: an IrMn (5 nm)/NiFe (4 nm) bilayer and a NiFe (4 nm) single layer (Figure R5), where the same current density flowing in the NiFe layer is used. This shows that there is no significant difference in PHE values between the two samples, demonstrating negligible contribution of the IrMn layer to the R_H of the IrMn/NiFe sample.

We included the above result as Supplementary Figure S2 in the revised manuscript.

Figure R5 | Planar Hall effect (PHE) of an IrMn (5 nm)/NiFe (4 nm) bilayer and a NiFe (4 nm) single layer. The reading current is 1 mA (0.83 mA) for the IrMn/NiFe bilayer (NiFe layer).

4. Why does the polycrystalline single NiFe layer have an easy axis as mentioned in line 128? I assume this is geometrically induced parallel to the Hall bar. Please mention this explicitly.

Response) The easy axis of the NiFe layer is developed when sputtered under an external magnetic field of 15 mT. Figure R6 (Fig. S1 of the original Supplementary Information) shows the magnetization curves as a function of magnetic fields along the x - and y -directions, demonstrating the easy axis of the NiFe layer along the x -direction, which is the magnetic field direction during sputtering. We included the following sentence in Methods section of the revised manuscript “During the sputtering process at room temperature, a magnetic field of 15 mT was applied to induce uniaxial anisotropy of the NiFe layer”.

Figure R6 | Magnetization curves of the Ta (5nm)/NiFe (4 nm) sample. They are measured with magnetic fields along the x - (solid blue) and y - (open red) directions. The sample is grown by sputtering under an external magnetic field of 15 mT along the x -direction.

5. On line 136, a potential similarity to Néel SOT is discussed. This is misleading, as a NSOT requires a special crystallographic symmetry, which only CuMnAs and Mn₂Au show. Thus this mechanism is not possible for IrMn.

Response) Thank you for the comment. We agree that the Néel SOT is not relevant in our polycrystalline IrMn. Therefore, we removed the sentence discussing the possibility of Neel SOT in the revised manuscript.

6. On lines 182-185, the role of uncompensated moments at the interface is discussed. In this framework, references to older work on exchange bias such as Phys. Rev. B 35, 3679 would be helpful.

Response) We apologize the missing relevant reference (*Phys. Rev. B 35, 3679*), which has been added to the revised manuscript as reference 49.

[Reviewer #2]

The manuscript by Kang et al. reports electrical reversible manipulation of the exchange bias direction in IrMn/NiFe structure. The authors observe a change in the exchange bias direction of IrMn/NiFe by applying an electric current into the structure and the change is maintained after the current is returned to zero. From a systematic investigation using several structures with reversed stacking order, Ta cap, various IrMn thicknesses, and NiFe layer thicknesses, they attribute their observation to the spin current generated in the IrMn layer through the spin Hall effect and exerted on the interfacial uncompensated moment of IrMn. In addition, they demonstrate a controllability of the change in an analog (memristive) fashion with good scalability and conclude that the established scheme is promising for neuromorphic computing. The exploration of new functionalities of antiferromagnetic materials is an active field in the spintronics, recognized as the antiferromagnetic spintronics, and I think this study adds a new option to this field as it provides a new means to take advantage of the natures of antiferromagnets. Furthermore, as the authors claim, I think this work may be important from technological point of view, because the neuromorphic computing requires scalable and highly controllable memristive devices. In these regards, I think the results presented in this manuscript "potentially" deserves to be published in high impact journals such like the Nature Communications. Nevertheless, in my opinion, the current manuscript has several critical issues which need to be addressed satisfactorily for further consideration. The details are listed below:

Response) We appreciate the reviewer's comment of *"I think the results presented in this manuscript "potentially" deserves to be published in high impact journals such like the Nature Communications."* However, she/he addresses critical issues with the statement that *"the current manuscript has several critical issues which need to be addressed satisfactorily for further consideration."* We respond to the reviewer's comments with additional experiments described below, which hopefully convince her/him that our work satisfies the criteria of *Nature Communications*.

1. Terminology

The authors describe that the easy axis is electrically manipulated, but this sounds to me odd. The easy axis represents the energetically favorable direction of magnetic moments that is determined by the local atomic structures or the global shape of the magnet; for example, the

easy axis of Co is along the c axis and easy axis of bar magnet is along the longitudinal direction. Meanwhile, I believe the phenomena observed in this work does not accompany any change in atomic structures and, of course, the shape of the device. Instead, in my understanding, what they observe is just a change in the exchange bias direction. Therefore, I think the authors should reconsider the meaning of "easy axis" and revise the manuscript accordingly.

Response) We acknowledge the reviewer's comment that the "easy axis" is not an appropriate terminology, and therefore we replaced it with "exchange bias" in the revised manuscript.

2. Quantification of several effects

If the authors wish to present these results in high impact journal like Nature Communications, I think more quantitative investigations are mandatory. For example, the temperature rise due to the Joule heating and its comparison to the blocking temperature are essential because the observed phenomena should be strongly dependent on the temperature. I also think that quantification of the current distribution in each layer; Oersted field, spin-orbit torque (sign and magnitude of the spin Hall angle) of IrMn and Ta is necessary; otherwise, their interpretation cannot be supported well. The value of current density in addition to the value of current should be also presented because the current density is more relevant when one considers the underlying physics.

Response) We appreciate the reviewer's comments on the quantitative analyses. We performed additional experiments to demonstrate (i) the blocking temperature of an IrMn/NiFe structure, (ii) estimated temperature rise during the switching experiments, (iii) current-induced switching experiment at a low temperature, (iv) Oersted field caused by the injecting current, and (v) spin-orbit torques arising from IrMn and Ta as follows.

(i) Blocking temperature of an IrMn/NiFe structure

We first measure the blocking temperature of the IrMn (5 nm)/NiFe (4 nm) sample that is used for the measurements shown in Fig. 1 of the main text. Figures R7a,b shows magnetization curves measured at temperatures ranging from 100 K to 400 K. The temperature dependence of the exchange bias field (B_{EB}) that is determined by the hysteresis shift is plotted in Figure R7c, demonstrating that the B_{EB} appears up to 380 K and disappears at temperatures greater than 385 K. The blocking temperature at which the exchange bias vanishes is 385 K for

the IrMn (5 nm)/NiFe (4 nm) sample.

Figure R7 | Temperature dependence of exchange bias in the IrMn (5 nm)/NiFe (4 nm) sample. a-b, Magnetization loops at different temperatures of 100~340 K (a) and 360~400 K (b). **c,** Exchange bias field B_{EB} versus temperature. B_{EB} is determined by the hysteresis shift in Figure R7a-b.

(ii) Temperature rise during the switching experiments

We next estimate the sample temperature during the current pulse application by measuring the longitudinal resistance R_{xx} of the IrMn (5 nm)/NiFe (4 nm) sample using an oscilloscope. Figure R8a shows the time-dependent R_{xx} while applying current pulses I_P of different magnitudes ranging from 0.1 mA to 27 mA (equivalent to $8.4 \times 10^{11} \text{ A/m}^2$). Here, the pulse width is 30 μs , identical to that used in the switching experiments of the original manuscript. As shown in the inset, the R_{xx} increases with the current magnitude. Note that the different pulse shapes are due to the different pulse modes according to the current range of the current source used in our experiment (Keithley 6221). Figure R8b shows the change in R_{xx} , $\Delta R_{xx} [= R_{xx}(I_P) - R_{xx}(I_P = 0.1 \text{ mA})]$ as a function of the I_P^2 , indicating the ΔR_{xx} is primarily due to Joule heating by the pulse current. Then, we measure the temperature dependence of R_{xx} and plot ΔR_{xx} versus temperature in Fig. R8c. By comparing those two graphs (Figs. R8b,c), we estimate the sample temperature when applying a current pulse density J_P as shown in Figure R8d. This indicates that the sample temperature increases to $354 \pm 3.5 \text{ K}$ when the critical switching current $(7.4 \pm 0.28) \times 10^{11} \text{ A/m}^2$ is applied (Fig. 3b of the original manuscript). This is lower than the blocking temperature of the sample (385 K), indicating that the current-induced manipulation of the exchange bias in our experiments is not due to Joule heating.

We included the following sentences regarding the results of thermal effect on page 8 of the revised manuscript and Supplementary Note 6,

“We note that the sample temperature increases due to the current injection, but not above the blocking temperature (Supplementary Note 6), which may reduce the AFM anisotropy and thereby assists the rotation of the AFM moment.”

Figure R8 | Joule heating effects in the IrMn (5 nm)/NiFe (4 nm) sample. a, Time (t)-dependent resistance measurements using oscilloscope. Here, a current pulse I_p ranges from 0.1 mA to 27 mA and the pulse width is 30 μ s. Inset shows the magnification of the time-dependent resistance data. **b**, Change in longitudinal resistance R_{xx} (ΔR_{xx}) as a function of I_p^2 . The R_{xx} value of each temperature is obtained by averaging the data from $t = 20 \mu$ s to $t = 30 \mu$ s. **c**, Temperature-dependent ΔR_{xx} of the sample. **d**, Estimated sample temperature T as a function of current density J_p . The line represents a fitting curve of $T=300+ 98.4J_p^2$

(iii) Current-induced switching experiment at a low temperature

To further elucidate Joule heating effect, we examine the current-induced manipulation of the exchange bias direction of the IrMn (5 nm)/NiFe (4 nm) sample at low temperatures. We performed current-induced exchange bias switching experiment at temperatures between 100 K and 300 K using the same measurement protocols described in the main text. Figure R9a shows the switching results of R_H versus pulse current J_p , demonstrating that the switching behaviour for all measuring temperatures is similar to that measured at 300 K (Fig. 3b of the

original manuscript). By comparing the maximum R_H value of each temperature with planar Hall resistance measured under an external magnetic field of 200 mT (Fig. R9b), we extract the rotation angle of the exchange bias (φ_{EB}) for each temperature (Fig. R9c). This demonstrates that the current-induced modulation of φ_{EB} is obtained regardless of the measuring temperature down to 100 K, which is much lower than the blocking temperature. This result confirms that the Joule heating effect is not a possible origin of the current-induced manipulation of exchange bias.

We included the following sentence regarding the results of switching experiment at a low temperature on page 8 of the revised manuscript and Supplementary Note 7.

“Similar switching behaviour is obtained at low temperatures down to 100 K, much lower than the blocking temperature, confirming that Joule heating effect is not a possible origin of the current-induced manipulation of exchange bias (Supplementary Note 7).”

Figure R9 | Current-induced manipulation of the exchange bias at low temperatures. a, The Hall resistance (R_H) of the IrMn (5 nm)/NiFe (4 nm) sample measured after applying an in-plane current pulse J_P at various temperatures. **b,** Planar Hall resistance R_H versus azimuthal angle of a magnetic field (φ_B) of 200 mT at different temperatures. **c,** The rotating angle of the exchange bias φ_{EB} versus J_P curves at various temperatures. The arrows denote the sweeping direction of J_P .

(iv) Oersted field by the injecting current

We also check if Oersted field caused by injecting current affects the current-induced modulation of the exchange bias. In this regard, we first estimate the current distribution of each layer by measuring the resistance R_{xx} of the devices with different IrMn and NiFe thicknesses. Here, NiFe thickness t_{NiFe} ranges from 3 to 15 nm in IrMn (5 nm)/NiFe (t_{NiFe}) structures and IrMn thickness t_{IrMn} ranges from 5 to 25 nm in IrMn (t_{IrMn})/NiFe structures.

Figure R10a shows the $(1/R_{xx}) \cdot L/W$ values of the samples as a function of t_{NiFe} and t_{IrMn} , where L and W is the length and width of the Hall-bar device, respectively. From the results, we extract the resistivity of each layer; 316 $\mu\Omega\text{cm}$ for IrMn and 51.3 $\mu\Omega\text{cm}$ for NiFe layer. This indicates that in the NiFe (4 nm)/IrMn (5 nm) bilayers, 83.1% and 16.9% of the current flows through the NiFe and IrMn layers, respectively. Moreover, in the IrMn (5 nm)/NiFe (4 nm)/Ta (1.5 nm) sample, where the resistivity of Ta is 353.8 $\mu\Omega\text{cm}$, 16.2%, 79.5%, 4.3% of the current flows through the IrMn, NiFe, and Ta (1.5 nm) layers, respectively.

Taking the current distributions into account, we calculated an Oersted field acting on NiFe layer in the IrMn (5 nm)/NiFe (4 nm) and IrMn (5 nm)/NiFe (4 nm)/Ta (1.5 nm) samples [*Nat. Nanotech.* 12, 980-986 (2017)]. Figure R10b,c shows the Oersted fields along the y and z direction ($B^{\text{Oe},y}$ and $B^{\text{Oe},z}$) generated in the middle of the NiFe layer of the samples. Here, we use a current density of $7.4 \times 10^{11} \text{ A/m}^2$, which is the switching current density of the sample. It is found that $B^{\text{Oe},y}$ is ~ 0.68 mT in the IrMn (5 nm)/NiFe (4 nm) sample and ~ 0.55 mT in the IrMn/NiFe/Ta sample. This allows us to exclude the Oersted effect from the main cause of the current-induced modulation of the exchange bias for the following two reasons; first, a larger rotation angle of exchange bias φ_{EB} is observed in the IrMn/NiFe/Ta sample, where the current-induced Oersted field is smaller. Second, such a small Oersted field of less than 1 mT cannot make any noticeable change in the magnetization direction of the exchanged coupled IrMn/NiFe sample as demonstrated in Fig. R10d, where planar Hall resistance R_{H} measured while a rotating magnetic field of 1 mT remains almost constant as compared to that of 200 mT. Note that the \cos component of R_{H} for 1 mT is probably due to the polar angle induced by a slight misalignment of the sample.

We included the above results as Supplementary Note 8 in the revised manuscript.

Figure R10 | Current distribution and Oersted field. **a**, $(1/R_{xx}) \cdot (L/W)$ as a function of NiFe thickness t_{NiFe} or IrMn thickness t_{IrMn} in IrMn/NiFe bilayer structures. **b,c** Calculated Oersted fields along the y and z direction ($B^{\text{Oe},y}$ and $B^{\text{Oe},z}$) in the middle of NiFe layer of the IrMn (5 nm)/NiFe (4 nm) sample (**b**) and the IrMn (5 nm)/NiFe (4 nm)/Ta (1.5 nm) sample (**c**). Here, a current density is $7.4 \times 10^{11} \text{ A/m}^2$. **d**, Planar Hall resistance R_H of the IrMn/NiFe sample measured with external magnetic fields of 200 mT and 1 mT.

(v) Spin-orbit torques arising from Ta and IrMn

To quantitatively estimate the spin Hall angles of Ta and IrMn, we performed in-plane harmonic measurements of the Ta (5 nm)/NiFe (4 nm) and IrMn (5 nm)/NiFe (4 nm) samples [*Phys. Rev. B* 90, 224427 (2014)]. Figures R11 a,b show the 2nd harmonic Hall resistance ($R^{2\omega}$) versus azimuthal angle (φ) curves of Ta/NiFe and IrMn/NiFe samples, which are measured with an ac current by rotating the samples under different external magnetic fields (B_{ext}). $R^{2\omega}$ is expressed as

$$R^{2\omega}(\varphi) = \left\{ \left[R_{\text{AHE}} \left(\frac{B_{\text{DLT}}}{B_{\text{eff}}} \right) + R_{\text{VT}} \right] \cos\varphi + \frac{[2R_{\text{PHE}}(B_{\text{FLT}} + B_{\text{Oe}})]}{B_{\text{ext}}} (2\cos^3\varphi - \cos\varphi) \right\}, \quad (1)$$

where R_{AHE} and R_{PHE} are the anomalous Hall and planar Hall resistances, respectively;

B_{DLT} (B_{FLT}) is the damping-like (field-like) effective field; B_{eff} is the effective magnetic field including the demagnetization field and anisotropy field of FM; $R_{\nabla T}$ is the thermal contributions, and B_{Oe} is the current-induced Oersted field. Figures R11c,d show the $\cos\varphi$ component of $R^{2\omega}$ divided by R_{AHE} [$R_{\text{cos}\varphi}^{2\omega}/R_{\text{AHE}}$] as a function of $1/B_{\text{eff}}$. Since the B_{DLT} and the associated effective spin Hall angle ($\theta_{\text{SH}}^{\text{eff}}$) is represented by the slope of the $R_{\text{cos}\varphi}^{2\omega}/R_{\text{AHE}}$ versus $1/B_{\text{eff}}$ curves, the negative (positive) slope in the Ta/NiFe (IrMn/NiFe) sample indicates a negative (positive) $\theta_{\text{SH}}^{\text{eff}}$ of Ta (IrMn). The B_{DLT} is obtained to be -1.2 ± 0.4 mT (2.3 ± 0.5 mT) for the Ta/NiFe (IrMn/NiFe) sample at a current density of 1.0×10^{11} A/m². These values are comparable to the previously reported ones [*Phys. Rev. B* 90, 224427 (2014), *Nat. Nanotechnol.* 11, 878-884 (2016)]. We also plot the $(2\cos^3\varphi - \cos\varphi)$ component of $R^{2\omega}$ divided by $2R_{\text{PHE}}$ [$R_{2\cos^3\varphi-\cos\varphi}^{2\omega}/2R_{\text{PHE}}$] as a function of $1/B_{\text{ext}}$, of which slope represents the combination of the B_{FLT} and B_{Oe} . The extracted $B_{\text{FLT}} + B_{\text{Oe}}$ is 0.05 ± 0.06 mT (0.13 ± 0.01 mT) for the Ta/NiFe (IrMn/NiFe) sample at a current density of 1.0×10^{11} A/m². This demonstrates that the B_{FLT} of the two samples is negligibly small as compared to B_{DLT} .

We included the above results ‘‘Spin Hall angle of Ta and IrMn’’ as Supplementary Note 9 in the revised manuscript

Figure R11 | In-plane harmonic spin-orbit torque measurements in Ta/NiFe and IrMn/NiFe structures. **a,b** Azimuthal angle (φ) dependent 2nd harmonic Hall resistance, $R^{2\omega}(\varphi)$, measured under a different B_{ext} in the Ta (5 nm)/NiFe (4 nm) (**a**) and IrMn (5 nm)/NiFe (4 nm) (**b**) samples. **c,d**, The extracted φ -dependent components of $R_{xy}^{2\omega}$; $\cos\varphi$ component versus $1/B_{\text{eff}}$ (**c**), $(2\cos^3\varphi - \cos\varphi)$ component versus $1/B_{\text{ext}}$ (**d**).

3. Physical scenario

I agree that the inference of the authors that "the obtained result originate from a spin current generated in IrMn and exerted on uncompensated moment of IrMn" is one of the possible scenarios, but I am wondering if this is the only one. I would suggest that the authors should consider other effects such as the spin anomalous Hall effect of NiFe and interfacial phenomena including the Rashba-Edelstein effect. Also, I think the spin torque acting on the magnetic moment of NiFe near the interface as well as on the uncompensated moment of IrMn should also play a certain role and cannot be excluded.

Response) We appreciate the reviewer's comment on possible origins of the current-induced manipulation of the exchange bias direction in IrMn/NiFe structures other than the spin current generated in IrMn. First of all, we cannot exclude the Rashba-Edelstein effect (REE) originated from the IrMn/NiFe interface since the REE contribution is difficult to distinguish from that of the bulk spin Hall effect due to the same symmetry. However, Fig. R11d shows that the B_{FLT} ,

governed primarily by the REE, is very small in our samples, indicating less contribution from the REE effect than that of the spin Hall effect. Second, the spin anomalous Hall effect can also generate a spin current, of which spin polarization is parallel to the magnetization direction [*Phys. Rev. B* 101, 134417 (2020)]. In order for the spin anomalous Hall effect generated from NiFe to exert torques on the uncompensated moments of IrMn, the magnetization direction of NiFe should be different from that of the IrMn uncompensated moments. However, it is not the case; the exchange bias of the IrMn/NiFe structures manifests that the two magnetization directions are aligned in parallel to each other. Therefore, we exclude the spin anomalous Hall effect as a possible cause. Finally, we would like to mention that the rotation angle of the exchange bias (ϕ_{EB}) is independent of NiFe thicknesses up to 10 nm (Fig. 3e in the main manuscript). This result suggests that the SOT-induced AFM switching is an interface phenomenon, i.e. the spin current gives spin torques to the interfacial uncompensated AFM moments. If spin currents generated in IrMn exert SOTs on NiFe, the magnitude of SOT (the rotation angle of the exchange bias) should decrease with an increase in t_{NiFe} . This is because spin currents are mostly absorbed at the FM interface when injected into an FM layer. With such reasons, we conclude that the spin current generated from IrMn layer give torques to the uncompensated AFM moments, and subsequently rotates the magnetization of the exchange-coupled IrMn/NiFe bilayers.

4. Comparison to previous study

I think the significance of this manuscript will be even improved by adding comparative discussion to previous related studies on antiferromagnet-ferromagnet structures after presenting their results. In particular, this work is similar to Ref. 26 (Lin et al., Nature Mater. 2019) where the electrical manipulation of exchange bias by SOT is reported although the direction of the exchange bias is different from the present manuscript. As far as I understand, explicit explanation is not given in Ref. 26 and I am wondering if the authors can provide some speculation to the underlying mechanism for the results reported in Ref. 26.

Response) We acknowledge that our work is similar to that of Ref. 26 [*Nat. Mater.* 18, 335-341 (2019)], in which the exchange bias is manipulated by an electrical current, and SOTs play a critical role in controlling the direction of the exchange bias. We believe that the mechanism proposed by our manuscript also occurs in their samples. However, in Ref. 26 the authors employ HM/FM/AFM trilayers, where spin currents are generated not only from the AFM layer

but also from the HM layer, therefore, the combination of the two spin currents will determine the overall SOT and switch the magnetization direction of the exchange-biased structure. Since it is difficult to quantitatively separate the contribution of the spin current from the AFM layer in the HM/FM/AFM trilayers, we cannot provide an explanation of the underlying mechanism for the results of Ref. 26 in the current manuscript. Further study is required to elucidate the SOT phenomena in the trilayer.

5. Information of the studied material system

Information of the studied material system is not sufficiently presented. At least, the following information should be presented so that the readers can perform replication study: composition of IrMn and NiFe, temperature of the stage during the deposition, underlayer (if any). In addition, the crystallographic structure, crystal orientation, and nanoscale texture of IrMn are also essential information because these factors are known to strongly relate to the exchange bias. Description on the possible magnetic structure of IrMn should be also added because it is known to crucially affect the spin transport properties of IrMn. In fact, it is reported that IrMn₃ generates out-of-plane component of spin accumulation [<https://doi.org/10.1103/PhysRevApplied.12.064046>], whereas such effect is not discussed in this manuscript.

Response) We apologize for the missing information on the materials used in our study. They are included in the revised manuscript as follows.

1. We add the sentence in Method section: “*During the sputtering process at room temperature, a magnetic field of 15 mT was applied to induce uniaxial anisotropy of the NiFe layer.*” Note that we don’t use any underlayer for the film growth.
2. We include the film composition of the Ir₂₅Mn₇₅ and Ni₇₆Fe₂₄ in Method section. The composition was measured using an IrMn (5 nm)/NiFe (4 nm) sample with inductively-coupled plasma optical emission spectroscopy (ICP-OES). Figures R12a-d show the intensities of Ir, Mn, Ni, and Fe along with the calibration line, which correspond to 0.093 mg/kg, 0.078 mg/kg, 0.074 mg/kg and 0.022 mg/kg, respectively. By considering the atomic mass of each material, we obtained the atomic composition of the deposited films.
3. We mention the crystalline texture of the IrMn layer on page 5 in the revised manuscript. “*We fabricate a (111) textured polycrystalline IrMn (5 nm)/NiFe (4 nm) layer by sputtering*

deposition". Figure R13 shows X-ray diffraction (XRD) patterns for the IrMn (15 nm)/NiFe (4 nm)/MgO (3.2 nm)/Ta (2 nm) sample deposited on the Si substrate, demonstrating two peaks at 33° and 41.3° , corresponding to the IrMn (111) plane and the Si (200) plane, respectively. Note that the out-of-plane component of spin polarization is only generated in (100)-textured IrMn [*Phys. Rev. Appl.* 12, 064046 (2019)], so it is not relevant for our samples with (111)-textured IrMn.

We included the above results of ICP-OES and XRD measurements in Method section and Supplementary Note 1 in the revised manuscript.

Figure R12 | Inductively-coupled plasma optical emission spectroscopy (ICP-OES) of the IrMn (5 nm)/NiFe (4 nm) sample. a-d, ICP-OES intensities of Ir (a), Mn (b), Ni (c), and Fe (d). Blue squares and lines denote the reference intensity and a linear fitting line, respectively.

Figure R13 | X-ray diffraction pattern of the IrMn (15 nm)/NiFe (4 nm)/MgO (3.2 nm)/Ta (2 nm) sample deposited on a Si substrate.

6. "memu" is used as a unit of magnetization curve but this is not suitable because, firstly, emu is not a unit of magnetization, secondly, the quantity with the unit of emu depends on the sample size whereas the information of the sample size is not described, and thirdly this manuscript adopts SI or MKSA unit system whereas "emu" is a unit of cgs system.

Response) We revised the unit of “memu” to MA/m in the revised manuscript. Figure R14 (Fig. 1b of the original manuscript) is an example of the modification.

Figure R14 | Hysteresis loop of IrMn (5 nm)/NiFe (4 nm) structure measured with magnetic fields along the x -axis (solid blue) and y -axis (open red).

7. "inversely proportional to" in line 167 is not appropriate because the result seems not show

the relation of $y = 1/x$. This part should be rephrased.

Response) We rephrased “*inversely proportional to*” to “*decreases with increasing the magnitude of B_{EB}* ” on page 9 of the revised manuscript.

8. "coherently" is used several times without any definition and I could understand its meaning after going through the manuscript. However, I think the meaning should be clearly defined somewhere.

Response) We apologize the ambiguous word “coherently, which we meant that the AFM moments of all domains rotate altogether. We want to distinguish it from the previous results in which the AFM moment is rotated by domain wall motion. However, as the reviewer pointed out, this is not necessary and can be confusing. Therefore, we rephrased it to “*collectively*” on page 3 (line 56) of the revised manuscript.

Reviewer #3 (Remarks to the Author):

This work reports a repeatable change of the exchange bias direction driven by electric current. This change seems to be reversible and it leads to a memristive behaviour that is also present in sub-micron wide stripes. The authors claim that all is due to spin orbit torques producing a rotation of the AFM easy axis (and/or amplitude), which makes the finding potentially quite interesting. On the other hand, the origin of the effect is unclear to me and the authors need to provide more measurements to build a stronger case and a stronger discussion.

Response) We appreciate the reviewer's comment that "*which makes the finding potentially quite interesting.*" However, she/he points out that "*the origin of the effect is unclear to me and authors need to provide more measurements to build a stronger case and a stronger discussion.*" We respond to the reviewer's comments with additional experiments described below, which will hopefully alleviate the reviewer's concerns.

A.- For instance, could the effect be the result of a combination of temperature plus Oersted field from the current achieving some sort of local annealing? The current densities are quite large, even larger than 10^{12} A/m² in the 500nm device. What happens for instance in Figure 1d if they go beyond 30 mA? The effect does not seem saturated. If they cannot go beyond 30 mA, because they destroy the sample, maybe this is an indication of a very relevant Joule heating. The authors acknowledge there may be some heating in line 149. In figure 2b they show that the effect grows when they have a 1.5nm Ta layer on top, claiming that the Ta layer diminishes the Oe field. I presume they say that because there is less current density. But, assuming the Ta is not oxidized (they mention a MgO/Ta capping layers, that I guess it is deposited in the same sputtering run), more current will flow through the Ta and NiFe layer than on the IrMn and the Oe field in the NiFe will be more intense. Fig.2b is an interesting result but does not completely rule out the possibility of the whole effect being simply current induced annealing.

In order to rule out this possibility, it is not enough with a simulation of a quick calculation following a theoretical model. The real sample may have a large interface thermal resistance with the substrate and the temperature could be larger than what any simulation would show. As the authors are using pulses in the microsecond range (which in terms of heating is pretty

much like a DC current) and they do not need transmission lines to apply the pulses, perhaps the best solution is to repeat (at least) one of the measurements on a cryostat, even if it is only at 77 K. If the effect is still there, it would give confidence that it is not a current induced annealing of the sample.

Response) We appreciate the reviewer for this critical comment. To rule out thermal effects as a possible cause of the results, we performed additional experiments: (i) current-induced switching experiment at a low temperature, (ii) measurement of the blocking temperature of an IrMn/NiFe structure, (iii) estimation of the temperature rise during the switching experiments, and (iv) calculation of Oersted field by the injecting current. As demonstrated below, the experimental results corroborate that the current-induced switching of the exchange bias is not due to thermal effects.

(i) Current-induced switching experiment at a low temperature

As suggested by the reviewer, we examine the current-induced manipulation of the exchange bias direction of the IrMn (5 nm)/NiFe (4 nm) sample at low temperatures. We performed current-induced exchange bias switching experiment at temperatures between 100 K and 300 K using the same measurement protocols described in the main text. Figure R15a shows the switching results of R_H versus pulse current J_P , demonstrating that the switching behaviour for all measuring temperatures is similar to that measured at 300 K (Fig. 3b of the original manuscript). By comparing the maximum R_H value of each temperature with planar Hall resistance measured under an external magnetic field of 200 mT (Fig. R15b), we extract the rotation angle of the exchange bias (φ_{EB}) for each temperature (Fig. R15c). This demonstrates that the current-induced modulation of φ_{EB} is obtained regardless of the measuring temperature down to 100 K, which is much lower than the blocking temperature. This result confirms that the Joule heating effect is not a possible origin of the current-induced manipulation of exchange bias.

We included the following sentence regarding the results of switching experiment at a low temperature on page 8 of the revised manuscript and Supplementary Note 7.

“Similar switching behaviour is obtained at low temperatures down to 100 K, much lower than the blocking temperature, confirming that Joule heating effect is not a possible origin of the current-induced manipulation of exchange bias (Supplementary Note 7).”

Figure R15 | Current-induced manipulation of the exchange bias at low temperatures. a, The Hall resistance (R_H) of the IrMn (5 nm)/NiFe (4 nm) sample measured after applying an in-plane current pulse J_P at various temperatures. **b,** Planar Hall resistance R_H versus azimuthal angle of a magnetic field (φ_B) of 200 mT at different temperatures. **c,** The rotating angle of the exchange bias φ_{EB} versus J_P curves at various temperatures. The arrows denote the sweeping direction of J_P .

(ii) Blocking temperature of an IrMn/NiFe structure

To further check thermal effect, we measured the blocking temperature of the IrMn (5 nm)/NiFe (4 nm) sample that is used for the measurements shown in Fig. 1 of the main text. Figures R16a,b shows magnetization curves measured at temperatures ranging from 100 K to 400 K. The temperature dependence of the exchange bias field (B_{EB}) that is determined by the hysteresis shift is plotted in Figure R16c, demonstrating that the B_{EB} appears up to 380 K and disappears at temperatures greater than 385 K. The blocking temperature at which the exchange bias vanishes is 385 K for the IrMn (5 nm)/NiFe (4 nm) sample.

Figure R16 | Temperature dependence of exchange bias in the IrMn (5 nm)/NiFe (4 nm) sample. a-b, Magnetization loops at different temperatures of 100~340 K (a) and 360~400 K (b). **c,** Exchange bias field B_{EB} versus temperature. B_{EB} is determined by the hysteresis shift in Figure R16a-b.

(iii) Temperature rise during the switching experiments

We then estimate the sample temperature during the current pulse application by measuring the longitudinal resistance R_{xx} of the IrMn (5 nm)/NiFe (4 nm) sample using an oscilloscope. Figure R17a shows the time-dependent R_{xx} while applying current pulses I_P of different magnitudes ranging from 0.1 mA to 27 mA (equivalent to $8.4 \times 10^{11} \text{ A/m}^2$). Here, the pulse width is 30 μs , identical to that used in the switching experiments of the original manuscript. As shown in the inset, the R_{xx} increases with the current magnitude. Note that the different pulse shapes are due to the different pulse modes according to the current range of the current source used in our experiment (Keithley 6221). Figure R17b shows the change in R_{xx} , $\Delta R_{xx} [= R_{xx}(I_P) - R_{xx}(I_P = 0.1 \text{ mA})]$ as a function of the I_P^2 , indicating the ΔR_{xx} is primarily due to Joule heating by the pulse current. Then, we measure the temperature dependence of R_{xx} and plot ΔR_{xx} versus temperature in Fig. R17c. By comparing those two graphs (Figs. R17b,c), we estimate the sample temperature when applying a current pulse density J_P as shown in Figure R17d. This indicates that the sample temperature increases to $354 \pm 3.5 \text{ K}$ when the critical switching current $(7.4 \pm 0.28) \times 10^{11} \text{ A/m}^2$ is applied (Fig. 3b of the original manuscript). This is lower than the blocking temperature of the sample (385 K), indicating that the current-induced manipulation of the exchange bias in our experiments is not due to Joule heating.

We included the following sentences regarding the results of thermal effect on page 8 of the revised manuscript and Supplementary Note 6,

“We note that the sample temperature increases due to the current injection, but not above the blocking temperature (Supplementary Note 6), which may reduce the AFM anisotropy and

thereby assists the rotation of the AFM moment.”

Figure R17 | Joule heating effects in the IrMn (5 nm)/NiFe (4 nm) sample. a, Time (t)-dependent resistance measurements using oscilloscope. Here, a current pulse I_P ranges from 0.1 mA to 27 mA and the pulse width is $30 \mu\text{s}$. Inset shows the magnification of the time-dependent resistance data. **b,** Change in longitudinal resistance R_{xx} (ΔR_{xx}) as a function of I_P^2 . The R_{xx} value of each temperature is obtained by averaging the data from $t = 20 \mu\text{s}$ to $t = 30 \mu\text{s}$. **c,** Temperature-dependent ΔR_{xx} of the sample. **d,** Estimated sample temperature T as a function of current density J_P . The line represents a fitting curve of $T=300+ 98.4J_P^2$

(iv) Oersted field by the injecting current

We also check if Oersted field caused by injecting current affects the current-induced modulation of the exchange bias. In this regard, we first estimate the current distribution of each layer by measuring the resistance R_{xx} of the devices with different IrMn and NiFe thicknesses. Here, NiFe thickness t_{NiFe} ranges from 3 to 15 nm in IrMn (5 nm)/NiFe (t_{NiFe}) structures and IrMn thickness t_{IrMn} ranges from 5 to 25 nm in IrMn (t_{IrMn})/NiFe structures. Figure R18a shows the $(1/R_{xx}) \cdot L/W$ values of the samples as a function of t_{NiFe} and t_{IrMn} , where L and W is the length and width of the Hall-bar device, respectively. From the results, we extract the resistivity of each layer; 316 $\mu\Omega\text{cm}$ for IrMn and 51.3 $\mu\Omega\text{cm}$ for NiFe layer. This indicates that in the NiFe (4 nm)/IrMn (5 nm) bilayers, 83.1% and 16.9% of the current flows through the NiFe and IrMn layers, respectively. Moreover, in the IrMn (5 nm)/NiFe (4 nm)/Ta (1.5 nm) sample, where the resistivity of Ta is 353.8 $\mu\Omega\text{cm}$, 16.2%, 79.5%, 4.3% of the current flows through the IrMn, NiFe, and Ta (1.5 nm) layers, respectively.

Taking the current distributions into account, we calculated an Oersted field acting on NiFe layer in the IrMn (5 nm)/NiFe (4 nm) and IrMn (5 nm)/NiFe (4 nm)/Ta (1.5 nm) samples [Nat. Nanotech. 12, 980-986 (2017)]. Figure R18b,c shows the Oersted fields along the y and z direction ($B^{\text{Oe},y}$ and $B^{\text{Oe},z}$) generated in the middle of the NiFe layer of the samples. Here, we use a current density of $7.4 \times 10^{11} \text{ A/m}^2$, which is the switching current density of the sample. It is found that $B^{\text{Oe},y}$ is ~ 0.68 mT in the IrMn (5 nm)/NiFe (4 nm) sample and ~ 0.55 mT in the IrMn/NiFe/Ta sample. This allows us to exclude the Oersted effect from the main cause of the current-induced modulation of the exchange bias for the following two reasons; first, a larger rotation angle of exchange bias φ_{EB} is observed in the IrMn/NiFe/Ta sample, where the current-induced Oersted field is smaller. Second, such a small Oersted field of less than 1 mT cannot make any noticeable change in the magnetization direction of the exchanged coupled IrMn/NiFe sample as demonstrated in Fig. R18d, where planar Hall resistance R_{H} measured while a rotating magnetic field of 1 mT remains almost constant as compared to that of 200 mT. Note that the \cos component of R_{H} for 1 mT is probably due to the polar angle induced by a slight misalignment of the sample.

We included the above results as Supplementary Note 8 in the revised manuscript.

Figure R18 | Current distribution and Oersted field. a, $(1/R_{xx}) \cdot (L/W)$ as a function of NiFe thickness t_{NiFe} or IrMn thickness t_{IrMn} in IrMn/NiFe bilayer structures. **b,c** Calculated Oersted fields along the y and z direction ($B^{\text{Oe},y}$ and $B^{\text{Oe},z}$) in the middle of NiFe layer of the IrMn (5 nm)/NiFe (4 nm) sample (**b**) and the IrMn (5 nm)/NiFe (4 nm)/Ta (1.5 nm) sample (**c**). Here, a current density is $7.4 \times 10^{11} \text{ A/m}^2$. **d,** Planar Hall resistance R_H of the IrMn/NiFe sample measured with external magnetic fields of 200 mT and 1 mT.

B.- My other big concern is that they are inferring a change in the exchange bias direction in the whole length of the strip, by measuring the PHE, which is only sensitive to what is happening in the Hall cross. Could they be measuring a local change that is happening only at the cross and not in the rest of the stripe. The authors should supply also AMR measurements (along the length of the strip), perhaps even using the existing longitudinal contacts if they do not want to process new Hall bars. It would be useful to contrast with an AMR hysteresis loop that the angle they are measuring with PHE is actually a change of the exchange bias direction along the entire nanostrip.

Response) We appreciate the reviewer's comment on the AMR measurement, which we did not consider in the original manuscript. As suggested by the reviewer, we performed AMR measurements of the IrMn (5 nm)/NiFe (4 nm) sample, in which the exchange bias direction is

initialized by an in-plane current pulse as $\varphi_{EB} = 0^\circ$, $\varphi_{EB} = +15^\circ$, and $\varphi_{EB} = -15^\circ$. The AMR curve along B_x clearly present that the center of the AMR curve of the sample with $\varphi_{EB} = 0^\circ$ shifts in the negative field direction (Fig. R19a), demonstrating an exchange bias along the positive x -direction. For the samples with $\varphi_{EB} = \pm 15^\circ$, we also observe shifts in the negative field direction, but its magnitude is smaller than that with $\varphi_{EB} = 0^\circ$, which is due to the reduced x -component of the exchange bias for the samples with $\varphi_{EB} = \pm 15^\circ$. Next, we measure the AMR while sweeping a magnetic field transverse to the current direction B_y (Figure R19b). No shift is observed when $\varphi_{EB} = 0^\circ$, which is expected because the exchange bias is developed in the x -direction. On the other hand, we find that the AMR loop shifts in the opposite directions: negative (positive) when $\varphi_{EB} = +15^\circ$ ($\varphi_{EB} = -15^\circ$), demonstrating that the y -component (or rotation angle) of the exchange bias field has the opposite sign. Furthermore, the AMR results are consistent with that of PHE measurement (Fig. 1 of the main text), confirming that the current-induced control of the exchange bias occurs over the whole length of the sample.

We included the following sentences regarding the AMR experiments on page 7 of the revised manuscript and Supplementary Note 2.

“We also measure the AMR effect of the bilayers, whose hysteresis loop shifts according to the φ_{EB} (Supplementary Note 2). This is consistent with those of PHE, demonstrating that the current-induced rotation of exchange bias occurs in the entire sample.”

Figure R19 | AMR measurement with different exchange bias directions. a,b, AMR measurements along the x -direction (a) and y -direction (b) of the IrMn (5 nm)/NiFe (4 nm) sample with a dc reading current of $100 \mu\text{A}$. The line and symbols indicate the different samples with $\varphi_{\text{EB}} = 0^\circ$ (red lines), $\varphi_{\text{EB}} = +15^\circ$ (blue open squares), $\varphi_{\text{EB}} = -15^\circ$ (blue solid squares). Schematics describe the AMR measurements of the samples with different φ_{EB} 's while sweeping magnetic fields along B_x (left) and B_y for (right). The solid arrows indicate the direction of the B_{EB} and the dotted arrows denote the x - (left) and y - (right) component of the B_{EB} . Here the initial φ_{EB} of $\pm 15^\circ$ is set by a current pulse of $\mp 8.4 \times 10^{11} \text{A/m}^2$.

C.- Being the IrMn such a popular material, it is incredible that nobody has seen this effect before. And there are plenty experiments with similar current densities in spin valves with IrMn. Do the authors find the same behaviour using other ferromagnetic material on top of the IrMn, such as FeCo for instance?

Response) We appreciate the reviewer's comment, which makes us investigate other exchange bias systems such as IrMn/FM (CoFeB, CoFe, Ni) structures. Figure R20a shows the hysteresis loop of the IrMn (13 nm)/CoFeB (4 nm) sample, demonstrating the exchange bias. Note that the exchange bias of the IrMn/CoFeB (4nm) structure is not formed when IrMn thickness is smaller than 13 nm. Figure R20b shows the planar Hall resistance (R_{H}) of the sample measured

with an external magnetic field of 200 mT. We then performed current-induced SOT switching experiments using the same measurement procedure of the main text (Fig. R20c). The result shows a similar switching behaviour to that of the IrMn/NiFe structure, but with a smaller rotation angle ($\varphi_{EB} = \pm 3^\circ$). The small effect may be due to the thick IrMn layer requiring a larger current to generate the SOT. We also find similar results in IrMn (12 nm)/CoFe (4nm) and IrMn (5 nm)/Ni (4 nm) structures (Figs. R21-R22), which exhibits $\varphi_{EB} = \pm 9^\circ$ and $\varphi_{EB} = \pm 8^\circ$, respectively. These results confirm that the SOT-induced manipulation of exchange bias generally occurs in IrMn/FM exchange-biased structures.

We included the following sentences regarding the results of switching experiment at a low temperature on page 7 of the revised manuscript and Supplementary Note 4.

“This is also observed in other IrMn/FM structures with various FMs of CoFe, CoFeB, and Ni. (Supplementary Note 4).”

Figure R20 | SOT-induced exchange bias switching for the IrMn/CoFeB structures. a, Hysteresis loop measured with a magnetic field along the x -axis (B_x) in the IrMn (13 nm)/CoFeB (4 nm) sample. **b,** R_H versus azimuthal angle of a magnetic field (φ_B) of 200 mT. **c,** The R_H vs J_P curves of the IrMn/CoFeB sample, where the arrows denote the sweeping direction of J_P .

Figure R21 | SOT-induced exchange bias switching for the IrMn/CoFe structures. a, Hysteresis loop measured with a magnetic field along the x -axis (B_x) in the IrMn (12 nm)/CoFe (4 nm) sample. **b,** R_H versus azimuthal angle of a magnetic field (φ_B) of 200 mT. **c,** The R_H vs J_P curves of the IrMn/CoFe sample, where the arrows denote the sweeping direction of J_P .

Figure R22 | SOT-induced exchange bias switching for the IrMn/Ni structures. a, Hysteresis loop measured with a magnetic field along the x -axis (B_x) in the IrMn (5 nm)/Ni (4 nm) sample. **b,** R_H versus azimuthal angle of a magnetic field (φ_B) of 200 mT. **c,** The R_H vs J_P curves of the IrMn/Ni sample, where the arrows denote the sweeping direction of J_P .

D.-The authors suggest that a spin current arising from the IrMn layer exert SOTs on uncompensated antiferromagnetic moments at the interface and then rotate the antiferromagnetic moments” but where is the evidence that, if there is a spin current, it is acting on the uncompensated moments? At the most, this is a hypothesis.

In general, the authors seem to perceive the anisotropy of the IrMn and the Exchange Bias field as the same thing. For instance, in lines 66-68, the authors state “Fi_AFM diminishes with an increase in the IrMn thickness, indicating that the SOT-induced rotation of the AFM easy axis is hindered by the AFM anisotropy”- I don’t understand this phrase – and then continues “which increases with its thickness”. How do they know that the anisotropy of the IrMn

increases with thickness in their samples? In figure 3a, they see an increase of the EB field with thickness of the IrMn, but I do not see why that may be a direct indication of the strength of the anisotropy. Thicker films should lead to a larger volume of the activated grains in the IrMn, but I do not see the straight link with the anisotropy.

In fact, throughout the article, the claim is the change of the anisotropy axis in the IrMn with the electric current, but the observation is a change in the direction of the EB with the electric current and I am not sure it is the same thing. They could be triggering a rotation of pinned rotatable uncompensated moments only at the interface (as they say in the abstract) or annealing with current some of the IrMn grains. Therefore, in my view, throughout the paper, they should speak about current induced persistent rotation of the Exchange Bias direction.

Response) We appreciate the reviewer's comment. To understand the mechanism of the current-induced control of the exchange bias direction, we examine its thickness dependence. First, we observe that the rotating angle of exchange bias direction (φ_{EB}) decreases with an increase in the IrMn thickness (t_{IrMn}) [Fig. 3c of the main text]. This attributed to the enhancement of the AFM anisotropy energy for a thicker t_{IrMn} [*Phys. Rev. B* 35, 3679 (1987); *J. Magn. Magn. Mater.* 192, 203–232 (1999); *Phys. Rev. B* 81, 212404 (2010)]. This is also confirmed by the increase in magnitude of the exchange bias for a thicker t_{IrMn} , which is determined by the AFM anisotropy energy. Therefore, the large AFM anisotropy can be responsible for the small rotating angle of the sample with a thick IrMn. Second, we observe that the current-induced control of the φ_{EB} is independent of the NiFe thicknesses up to 10 nm (Fig. 3f of the main text). This allows us to exclude the mechanism that spin currents generated in IrMn directly exert SOTs on NiFe, in which case the magnitude of SOT (or the maximum φ_{EB} value) should decrease with an increase in t_{NiFe} . This result suggests that the SOT-induced exchange bias switching is an interface phenomenon; therefore, we infer that the spin current gives spin torques to the interfacial uncompensated AFM moments, rotating the magnetization directions of the exchange-coupled IrMn and NiFe simultaneously.

Note that we acknowledge the reviewer's comment that the control of the “AFM easy axis” may be misleading. Therefore, we changed it to “exchange bias direction” in the revised manuscript.

E.-When they mention a current value (i.e 15 mA) they should give also the current density value, at least the first time they mention that value of current. In general, throughout the paper

is better to give also values of the current density.

Response) As suggested by the reviewer, we changed the current to current density in all figures and descriptions in the revised manuscript. Figure R23 is an example of a revised plot (Fig. 1d in the original manuscript) with current density values on the x-axis.

Figure R23 | The R_H vs J_P curves of the IrMn (5 nm)/NiFe (4 nm).

F.-In the experiment described in lines 145 to 147, where they grow the NiFe/IrMn, is not that the same as changing the direction of the current in the IrMn/NiFe sample? I do not quite see why is different and why it is an experiment that proves anything.

Response) We appreciate the reviewer's comment. We believe that there are two possible scenarios for the current-induced control of the exchange bias direction; first, spin currents arising from the IrMn give torques on the IrMn moments itself. Second, the spin current generated in IrMn induces spin accumulation at the interface, exerting torques on the NiFe moments or the uncompensated AFM moments. In the latter case, where opposite spins are accumulated on the top and bottom interfaces, the rotation direction will reverse when changing the stacking order, whereas it is independent of the stacking order in the former case. As shown in Fig. 2a, the switching polarity is reversed, which excludes the former scenario.

G.-In Figure 3b, can they measure to slightly higher current densities? It would also be nice (like in the rest of the figures) to show few points of the saturation of the effect and, in this case, plot all the curves with the same maximum and minimum current densities. My guess here is that the authors cannot plot all the curves with the same maximum current density because of the heating. For the same current density, thicker samples heat up more, and my guess is that

they may destroy the samples. Is this the case?

Response) As pointed out by the reviewer, we cannot further increase the current density for samples with thicker IrMn because they are broken by a high current density. Therefore, the rotating angle might be slightly underestimated for the samples with a thicker IrMn. Figure R24 shows a magnified image of the φ_{EB} versus J_P curves in the range from $0.5 \times 10^{12} \text{A/m}^2$ to $1 \times 10^{12} \text{A/m}^2$. Though the complete saturation is not observed, the slopes of the curves gradually decrease with increasing J_P . Therefore, it is reasonable to claim that the φ_{EB} decreases with the increase of IrMn thickness. We believe that we can improve the experimental conditions to increase the current density such as employing a substrate with better heat dissipation or a shorter current pulse.

Figure R24 | NiFe thickness dependence of the exchange bias switching (left) and the magnified version plotted from current density from $0.5 \times 10^{12} \text{A/m}^2$ to $1 \times 10^{12} \text{A/m}^2$ (right).

-H Figure 3e is quite puzzling for me. The fact that the effect does not depend at all with the thickness of the NiFe does not quite fit with current induced thermal annealing, but I don't quite understand it either from the point of view of a SOT. For a fixed exchange bias field, the thicker the NiFe layer, the larger the energy involved. So, how can the same sequence of current pulses deliver the same Exchange Bias rotation regardless of the thickness of the NiFe? If the authors understand it, they should elaborate a bit more their explanation. In line 183, they conclude that it is an interface phenomenon but then, it would decrease inversely proportional to the thickness of NiFe. In any case, I suggest to include two measurements:

H(a).- The results for NiFe thickness 15 and 20 nm. I guess at some point the thickness of the NiFe must be detrimental.

H(b).- This figure is one of the reasons I am suggesting, on top of the PHE measurements that

the article provide, AMR measurements to see the hysteresis loop of the entire length of the strip, and rule out that what the authors are seeing is just a local fluctuation of the magnetization in the cross.

Response) We thank you for the comments and suggestions. We performed the additional experiments as follows: experiment with a thicker NiFe and AMR measurement

As shown in Fig. 3e, we observe that the rotation angle of the exchange bias direction is $\sim 30^\circ$ for the samples regardless of NiFe thickness (t_{NiFe}) up to 10 nm. As the reviewer pointed out, the rotation angle of the exchange bias should decrease with increasing t_{NiFe} if SOT directly acts on the NiFe layer. Since this is not the case, we infer that the spin current gives spin torques to the interfacial uncompensated AFM moments, rotating the magnetization directions of the exchange-coupled IrMn and NiFe simultaneously. As the reviewer suggested, we performed experiments with t_{NiFe} of 15 nm and 20 nm. As shown in Fig. R25, the φ_{EB} reduces when t_{NiFe} is larger than 15 nm. This is attributed to the larger magnetic energy of thick NiFe, which is detrimental to the SOT-induced switching of the magnetization.

We performed AMR measurements of the IrMn/NiFe samples with different exchange bias directions while applying magnetic fields along the x - and y -directions. As described in the response to the query B above (Fig. R19), the AMR results confirm that the current-induced control of the exchange bias occurs over the whole length of the sample.

Figure R25 | Thickness dependence of SOT-induced exchange bias switching in IrMn/NiFe (t_{NiFe}) when $t_{\text{NiFe}} = 5 \sim 20$ nm. Here the arrows denote the sweeping direction of J_p .

1.-Title: As the claimed effect is achieved with electric current, perhaps the title should be “Current induced manipulation of...”

Response) As reviewer suggested, we changed the title of the revised manuscript to “*Current-induced manipulation of exchange bias in IrMn/NiFe bilayer structures.*”

2.-Line 16: “To date, the manipulation of antiferromagnetic moments...” perhaps should clarify that is manipulation by electric means or by electric current.

Response) We added the phrase “*by electric current*” on page 1 (line 16) of the revised manuscript.

3.-Line 19: “...electrical manipulation...” by “current induced manipulation”

Response) We changed “electrical manipulation” to “current-induced manipulation” on page 1 (line 19) of the revised manuscript.

4.-Paragraph from line 51 to 58 is very difficult to understand.

Response) We thank you for the comment and we revised the paragraph as follows.

Notably, it has been demonstrated that the electrical manipulation of AFM moment typically shows multi-level characteristics [*Nat. Commun.* 8, 15434 (2017), *Nat. Electron.* 3, 92-98 (2020), *Appl. Phys. Lett.* 110, 92410 (2017)]; the direction of the AFM moment can be gradually modulated by the magnitude and polarity of a writing current. However, the multi-level characteristics rely on the AFM domain structure because the current-induced SOT controls the overall AFM moments by switching the AFM moment in some domains and/or by driving the AFM domain wall [*Phys. Rev. Lett.* 118, 57701 (2017), *Adv. Func. Mater.* 30, 1909092 (2020), *Nat. Commun.* 11, 5715 (2020), *Appl. Phys. Lett.* 110, 92410 (2017)]. Therefore, to exploit the memristive behaviour in nano-devices, it is necessary to either engineer an AFM with nanometre-sized domains or find a way to control the entire AFM moment collectively.

5.-Line 63. “...and the associated exchange bias between up to ± 22 degrees”. What does this mean?

Response) We feel sorry for the unclear sentence. We modified it on page 3 of the revised manuscript. *“This demonstrates that the SOT caused by the spin Hall effect in IrMn effectively controls the exchange bias direction in a range of ± 22 degrees.”*

6.-Line 85. *“... by deposition” or “by sputtering deposition”?*

Response) We modified “deposition” to “sputtering deposition” on page 4 of the revised manuscript

7.-Figure 1d. *As in figure 1c, put the field B at which this measurement is taken. I guess $B=0$ mT.*

Response) We appreciate the reviewer’s comment. We added the B value ($B = 0$ mT) in Fig. R26 (Fig. 1d of the original manuscript) of the revised manuscript

Figure R26 | The R_H vs J_P curves of the IrMn (5 nm)/NiFe (4 nm) under no external magnetic field, where the arrows denote the sweeping direction of J_P .

8.-Figure 1d. *I think it is better to give the values of current in the x-axis as a current density.*

Response) As suggested by the reviewer, we changed the current to current density in all figures (including Fig. 1d) and descriptions of the revised manuscript.

9.-Line 110: *“..demonstrating the electrical modulation of ϕ_m of 15° in a reversible manner”.* *I don’t think ‘modulation’ is the right word. Consider something in the lines of ‘demonstrating*

a repeatable commutation of φ_m between $\pm 15^\circ$ with electric current’.

Response) As suggested by the reviewer, we revised the sentence as “a repeatable commutation of φ_m between $\mp 15^\circ$ with electric current” on page 6 of the revised manuscript.

10.-Line 121: “The spontaneous recovery...”, they mean the recovery when the field is reduced to zero, so it is not really spontaneous. Remove ‘spontaneous’?

Response) As suggested by the reviewer, we remove the word ‘spontaneous’ in line 123 of the revised manuscript.

11.-Line 161: In reference to Figure 3 it mentions that the thickness of IrMn goes from 5 to 25nm, but in the figure they only show 5, 10 and 15 nm. One has to go to supplementary material to see the 20 and 25 nm. I believe in delivering most of the important information in the main text, rather than in the Supplementary Information. Therefore, it would be nice if they could somehow include the 20 and 25 nm IrMn thickness in the figure 3b.

Response) As suggested by the reviewer, we included all data for different IrMn thicknesses from 5 to 25nm in Fig. R27a,b (Fig. 3a,b of the original manuscript). We note that 20 nm data was not included in our original manuscript

Figure R27 | Thickness dependence of SOT-induced exchange bias switching. a, Hysteresis loop measured using a magnetic field along the x -axis, B_x for samples with different t_{IrMn} 's ranging from 5 to 25 nm **b,** φ_{EB} versus current density (J_p) curves, where the arrows denote the sweeping direction of J_p for samples with different t_{IrMn} 's ranging from 5 to 25 nm

12.-Figure 4: Plot the x-axis in terms of current density.

Response) As suggested by the reviewer, we changed the current to current density in all figures (including Fig. 4) and descriptions in the revised manuscript.

13.-Line 201: “This implies...”. I do not see why the experiments of the memristive behaviour imply collective rotation of the AFM easy axis. Perhaps after all the corrections are addressed, this point may become clearer.

Response) We thank the reviewer for the comment. As the previous reports on memristive behaviour of the exchange bias switching are attributed to the AFM domain wall motion, multilevel characteristics gradually vanishes with the reduction of device size [*Appl. Phys. Lett.* 110, 092410 (2017)]. However, in our experiment, the multilevel switching characteristics are successfully achieved in a 500 nm-sized device, comparable to those in the 4 μm -sized device. With this result, we infer that the AFM moments and the exchange-coupled FM moments are rotated collectively by the current-induced SOT, which is distinct from the previous results based on AFM domain wall motions.

14.-Although I am not a native speaker myself, I think the manuscript requires a general English revision.

Response) We thank for the comment. In fact, the original manuscript has been proofread in English by the KAIST language center. However, as suggested by the reviewer, we revised the manuscript with the help of a native speaker.

Reviewers' Comments:

Reviewer #1:

Remarks to the Author:

In the revised version of their manuscript and in the associated reply letter, Kang et al. carefully addressed all questions, which I asked in my previous report. Based on additional experiments, they were able to resolve all my concerns regarding a potentially thermal origin of their observation. Also an issue concerning the terminology of anisotropy and exchange bias is now resolved.

Thus, I gladly recommend the publication of the manuscript in its present form.

Reviewer #2:

Remarks to the Author:

All of my questions in my previous review reports were clarified by the additional experiments and explanations described in their point-by-point response, and I would like to support the publication of this manuscript in Nature Communications. However, if I have not missed, the authors have not made any revisions for my previous comments of "3. Physical scenario". I think their argument in their response letter is reasonable, but it is not clear to the potential readers unless described in the main text. Thus, I recommend that they elaborate a bit more on the physical mechanism in the discussion section.

Reviewer #3:

Remarks to the Author:

The authors have answered to all the issues risen by the referees and now the experiment looks clearer. Sadly, with the new evidence, I still believe that Joule heating is a key player in the results and, at the very least, the discussion on Joule heating has to be revised and brought to the main text so the reader can evaluate it. It is not enough to write a sentence saying "Joule heating is not the reason, go to Suppl. Info.". Before going into the details, I would like to stress that the three referees were concerned about the same issues and Joule heating was perhaps the most important for all of us. In fact, while reading the ref. reports of all the referees, for a while, I was not sure which one was mine. This means that all of us spotted similar weaknesses that need to be addressed very well. I hope the authors understand that, for the purpose of publishing in nature, it is not the same that the effect is based on spin transfer or based (or even mostly based) on current induced thermal annealing. Therefore, this point has to be clarified meticulously.

Note that, even if all the values are correct (Temperature 354 K and around 1mT Oe field induced by the current), the Exchange Bias is notoriously reduced for that temperature (see Figure R1c) and it is gone for 380 K, so a difference of only 25 K in your estimation of the temperature can be a game changer. Also, on your estimation of the Oersted field created by the current, you conclude with figure R4d that 1mT does not make any change in comparison to the change obtained with 200 mT. This measurement though is performed at Room Temperature, while the 1 mT would be acting on a hot strip. Therefore, figure R4d is misleading.

Allow me first to introduce my discussion with some facts:

A.-Joule heating is proportional to the current density to the square, to the resistivity of the strip, to the thickness of the strip, proportional to the thickness of the substrate that dissipates the heat and inversely proportional to the thermal conductivity of that thickness of the substrate that dissipates the heat. For instance, if you have a Si/SiO₂ substrate, the important layer is the SiO₂ layer. All this information has to be given in the main text, so the reader can simulate the temperature of your system. I could not find information on the substrate and this is vital for the thermal evaluation. The resistivity of the layers is in the Suppl. Info., and they have to be brought to the main text when they build a discussion on Joule heating. Again, information on the substrate has to be given (ie. "... the samples were deposited on Si/SiO₂(100nm)" for instance... that is enough).

Note that the Joule heating is proportional to the thickness of the strip for a constant current density. You will notice in your experiments, the thicker the strip, the lower the current density you can apply without destroying the device. Also, if you go to Figure 3b&e of the revised version

of the manuscript, the thinner the IrMn, the larger the current density required to cycle the rotation of the EB. The same happens with the thickness of NiFe. This fits qualitatively with the fact that heating is proportional to the thickness of the strip for a constant current density.

B.-A metallic nano or micro strip can reach 600-700 K repeatedly without losing any of its structural properties. This is not a "stressful" temperature for the strip. On the other hand, for hotter temperatures, the strip quickly degrades and I have not been able to measure 1000 K on any strip before destruction. Usually 1012 A/m² marks the onset of serious heating. Therefore, if the samples get destroyed around 1012 A/m² (which it should be around 1000 K), I find very intriguing that the authors estimate a very mild increase of temperature of 380K for 0.8·1012 A/m².

C.-Although I was the one suggesting the measurement at low temperature, after doing some simulations, I was myself surprised to see that it does not rule out the Joule heating. If you look to figure R3a, the effect is present at low temperature, but it requires a larger current density (the loops are wider). It is not much, but this is precisely the range of current density where a small increase in current density implies a large temperature change because the strip can no longer dissipate so much heat to the substrate, even if the substrate is cold. To give you a figure, at 3·1012 A/m², there is nothing you can do to avoid destruction by heating, even if you measure at 4K.

D.-There are several papers that report changes on the exchange bias with temperature. Notoriously, not long ago [Nature Materials 17 No 1, 28-35 (2018)] a spontaneous structural change (driving a change in the exchange bias) was reported on IrMn at room temperature. The effect took place over several hours but it happened at room temperature. Therefore, if the microstructure of the IrMn is right, one should not rule out a rapid thermal annealing of the IrMn caused by the current density in a matter of microseconds.

With all these facts in mind, I believe the thermal analysis has to be more consistent. If, after this second round, the authors come back with a mild temperature increase, the thermal discussion has to go in the main text because, at the very least, it is playing a part in this effect. At least Figure R1c (Exchange Bias with temperature) and the important information of Figure R2, once improved (see below).

1.-Calibration of the temperature. Although it was me in the first place suggesting the calibration with the pulsed current and an oscilloscope, this was because I mistakenly thought the pulses were delivered with a nanosecond pulse generator (at constant voltage). I also was hoping that there would not be any relevant heating. The Keithley 6221 has the pulsed current feature precisely to remove the extra voltage caused by heating, in combination with the nanovoltmeter 2182A. To avoid any potential mistakes, I think the best option is to use a DC current. Using microsecond pulses is the same as using DC current as the temperature on a metallic strip rises in nanoseconds. Measure Rxx vs. temperature (the authors mention this plot but they do not include it. Measure Rxx vs current density (DC) and workout Temperature versus current density, as they have done, but with DC current. These three plots (Rxx vs T, Rxx vs J and T vs J) should be in the main text together with Figure R1c, exchange bias vs temperature.

2.- Describe the substrate in the main text, in particular the thickness of SiO₂ if it is a Si/SiO₂ substrate. While describing the figure of heating (see previous point) give values of resistivity of the layers, although the Rxx vs. Temperature (being Rxx the resistance of the strip) is perhaps the most valuable information in case someone wants to simulate it. If there is any capping layer that protects the devices (i.e. A 3nm TaOx layer deposited over the entire sample), that should be mentioned too.

3.- The Oersted field generated by the current density was calculated referring to an article Nat.Nanotech.12, 980-986 (2017), but at a glance I could not see the link. Ideally, they should calculate the field with a finite element simulation (i.e. COMSOL or similar). If not, provide the formula they are using or refer to it clearly (i.e. formula 3 or Ref X). Then Figure R4d should be re-done. The blue curve in that figure, has to be measured at the temperature they obtain with their thermal estimation and then we will be able to see how far does 1mT go at a larger temperature. This revised figure R4d, has to be brought to the main text, together with the figures described in point 1. It would be a figure dedicated to rule out Joule heating plus Oersted field current annealing.

4.- Line 67. The authors mention again that the anisotropy of the IrMn changes with thickness. How do they know that? The origin of the anisotropy is in the structure of the IrMn and it will unlikely change from one thickness to another if the deposition conditions are the same. Do they mean exchange bias energy?

Dear reviewers

We appreciate your comments and valuable queries, which have helped us improve the clarity and quality of our manuscript. Given below are detailed point-by-point responses to your questions and suggestions. The corresponding modifications are incorporated in the revised manuscript (marked in blue). We believe our manuscript has been improved significantly and now deserves publication in *Nature Communications*.

Yours sincerely,

Byong-Guk Park on behalf of all co-authors

[Reviewer #1]

In the revised version of their manuscript and in the associated reply letter, Kang et al. carefully addressed all questions, which I asked in my previous report. Based on additional experiments, they were able to resolve all my concerns regarding a potentially thermal origin of their observation. Also, an issue concerning the terminology of anisotropy and exchange bias is now resolved. Thus, I gladly recommend the publication of the manuscript in its present form.

Response) We thank the reviewer for carefully reading our responses. We are happy to hear that “*Based on additional experiments, they were able to resolve all my concerns regarding a potentially thermal origin of their observation. Also, an issue concerning the terminology of anisotropy and exchange bias is now resolved,*” and “*I gladly recommend the publication of the manuscript in its present form.*”

[Reviewer #2]

All of my questions in my previous review reports were clarified by the additional experiments and explanations described in their point-by-point response, and I would like to support the publication of this manuscript in Nature Communications. However, if I have not missed, the authors have not made any revisions for my previous comments of “3. Physical scenario”. I think their argument in their response letter is reasonable, but it is not clear to the potential readers unless described in the main text. Thus, I recommend that they elaborate a bit more on the physical mechanism in the discussion section.

Response) We appreciate the reviewer’s comments that “*I would like to support the publication of this manuscript in Nature Communications.*” Following the reviewer’s recommendation, we added the following paragraph regarding the physical scenario of the exchange bias switching on page 10 of the revised manuscript.

“We want to discuss other possible contributions that can generate spin torques to the current-induced manipulation of exchange bias direction, such as the Rashba-Edelstein effect (REE) of the IrMn/NiFe interface or the spin anomalous Hall effect (SAHE) of the NiFe layer. First, we cannot completely exclude the REE originated from the IrMn/NiFe interface due to the same symmetry as the bulk spin Hall effect. However, field-like SOT, governed primarily by the REE, is very small in our samples (Fig. S10), indicating less contribution from the REE effect than the spin Hall effect. Second, the SAHE can also generate a spin current, of which spin polarization is parallel to the magnetization direction⁴⁷. For the SAHE generated from NiFe to exert torques on the uncompensated moments of IrMn, the magnetization direction of NiFe must be different from the direction of the IrMn uncompensated moments. However, it is not the case; the exchange bias of the IrMn/NiFe structures manifests that the two directions are aligned in parallel to each other. Therefore, we exclude the SAHE as a possible cause.”

[Reviewer #3]

The authors have answered to all the issues risen by the referees and now the experiment looks clearer. Sadly, with the new evidence, I still believe that Joule heating is a key player in the results and, at the very least, the discussion on Joule heating has to be revised and brought to the main text so the reader can evaluate it. It is not enough to write a sentence saying “Joule heating is not the reason, go to Suppl. Info.”. Before going into the details, I would like to stress that the three referees were concerned about the same issues and Joule heating was perhaps the most important for all of us. In fact, while reading the ref. reports of all the referees, for a while, I was not sure which one was mine. This means that all of us spotted similar weaknesses that need to be addressed very well. I hope the authors understand that, for the purpose of publishing in nature, it is not the same that the effect is based on spin transfer or based (or even mostly based) on current induced thermal annealing. Therefore, this point has to be clarified meticulously.

Response) We appreciate the reviewer’s comment of “*The authors have answered to all the issues risen by the referees and now the experiment looks clearer.*” However, the reviewer pointed out that “*effects of current induced thermal annealing has to be clarified meticulously.*” We respond to the reviewer’s comments below with additional experiments suggested by the reviewer, which demonstrates that thermal effects are not the primary cause of the current-induced manipulation of exchange bias.

Note that, even if all the values are correct (Temperature 354 K and around 1mT Oe field induced by the current), the Exchange Bias is notoriously reduced for that temperature (see Figure R1c) and it is gone for 380 K, so a difference of only 25 K in your estimation of the temperature can be a game changer. Also, on your estimation of the Oersted field created by the current, you conclude with figure R4d that 1mT does not make any change in comparison to the change obtained with 200 mT. This measurement though is performed at Room Temperature, while the 1 mT would be acting on a hot strip. Therefore, figure R4d is misleading.

Response) We appreciate the reviewer’s important comment, and we agree that the planar Hall effect measurement should be conducted at a sample temperature where the switching occurs. As the reviewer suggested, we measured again the planar Hall effect at 354 K with an external magnetic field of 1 mT, of which magnitude is comparable to that measured at room

temperature. Experimental results are described in the response to the reviewer's comment #3 below.

Allow me first to introduce my discussion with some facts:

A.-Joule heating is proportional to the current density to the square, to the resistivity of the strip, to the thickness of the strip, proportional to the thickness of the substrate that dissipates the heat and inversely proportional to the thermal conductivity of that thickness of the substrate that dissipates the heat. For instance, if you have a Si/SiO₂ substrate, the important layer is the SiO₂ layer. All this information has to be given in the main text, so the reader can simulate the temperature of your system. I could not find information on the substrate and this is vital for the thermal evaluation. The resistivity of the layers is in the Suppl. Info., and they have to be brought the main text when they build a discussion on Joule heating. Again, information on the substrate has to be given (ie. "... the samples where deposited on Si/SiO₂(100nm)" for instance... that is enough).

Note that the Joule heating is proportional to the thickness of the strip for a constant current density. You will notice in your experiments, the thicker the strip, the lower the current density you can apply without destroying the device. Also, if you go to Figure 3b&e of the revised version of the manuscript, the thinner the IrMn, the larger the current density required to cycle the rotation of the EB. The same happens with the thickness of NiFe. This fits qualitatively with the fact that heating is proportional to the thickness of the strip for a constant current density.

B.-A metallic nano or micro strip can reach 600-700 K repeatedly without losing any of its structural properties. This is not a "stressful" temperature for the strip. On the other hand, for hotter temperatures, the strip quickly degrades and I have not been able to measure 1000 K on any strip before destruction. Usually 1012 A/m² marks the onset of serious heating. Therefore, if the samples get destroyed around 1012 A/m² (which it should be around 1000 K), I find very intriguing that the authors estimate a very mild increase of temperature of 380K for 0.8·1012 A/m².

C.-Although I was the one suggesting the measurement at low temperature, after doing some simulations, I was myself surprised to see that it does not rule out the Joule heating. If you look to figure R3a, the effect is present at low temperature, but it requires a larger current density (the loops are wider). It is not much, but this is precisely the range of current density where a small increase in current density implies a large temperature change because the strip can no longer dissipate so much heat to the substrate, even if the substrate is cold. To give you a figure,

at 3·10¹² A/m², there is nothing you can do to avoid destruction by heating, even if you measure at 4K.

Response to point A-C) We appreciate the reviewer's kind explanation on the thermal effects. We want to note that we used high-resistive Si substrates without SiO₂ on top, which was mentioned in the Method section of the original manuscript. We believe the mild temperature increase in our sample is due to a large thermal conductivity of the high-resistive Si substrate (See the response to the reviewer's comment #2 below). To alleviate the reviewer's concern, we measured again the current-induced temperature rise with DC current, which are described in the response to the reviewer's comment #1 below.

D.-There are several papers that report changes on the exchange bias with temperature. Notoriously, not long ago [Nature Materials 17 No 1, 28-35 (2018)] a spontaneous structural change (driving a change in the exchange bias) was reported on IrMn at room temperature. The effect took place over several hours but it happened at room temperature. Therefore, if the microstructure of the IrMn is right, one should not rule out a rapid thermal annealing of the IrMn caused by the current density in a matter of microseconds.

Response) We understand the reviewer's concern about the thermal annealing effect due to the current-induced Joule heating and Oersted field. Following to the reviewer's suggestions, we performed two additional measurements of (i) temperature increase due to the current-induced Joule heating using DC current and (ii) planar Hall effect at an elevated temperature where the current-induced switching occurs. As you can see in the responses to the reviewer's comments #1 and #2, the experimental results corroborate that the Joule heating effect is not a major cause of the current-induced modulation of exchange bias.

With all these facts in mind, I believe the thermal analysis has to be more consistent. If, after this second round, the authors come back with a mild temperature increase, the thermal discussion has to go in the main text because, at the very least, it is playing a part in this effect. At least Figure R1c (Exchange Bias with temperature) and the important information of Figure R2, once improved (see below).

Response) We appreciate the reviewer's comment. Following the reviewer's suggestion, we included the discussion about the thermal effects in the main text of the revised manuscript.

1. Calibration of the temperature. Although it was me in the first place suggesting the calibration with the pulsed current and an oscilloscope, this was because I mistakenly thought the pulses were delivered with a nanosecond pulse generator (at constant voltage). I also was hoping that there would not be any relevant heating. The Keithley 6221 has the pulsed current feature precisely to remove the extra voltage caused by heating, in combination with the nanovoltmeter 2182A. To avoid any potential mistakes, I think the best option is to use a DC current. Using microsecond pulses is the same as using DC current as the temperature on a metallic strip rises in nanoseconds. Measure R_{xx} vs. temperature (the authors mention this plot but they do not include it. Measure R_{xx} vs current density (DC) and workout Temperature versus current density, as they have done, but with DC current. These three plots (R_{xx} vs T , R_{xx} vs J and T vs J) should be in the main text together with Figure R1c, exchange bias vs temperature.

Response) We appreciate the reviewer for this critical comment, allowing us to re-examine the thermal effect and unambiguously demonstrate that this is not the primary cause of our results. As the reviewer suggested, we estimated a sample temperature rise with two methods of using DC current and the pulse delta mode of Keithley 6221 and 2182A. We first measured the longitudinal resistance R_{xx} of the IrMn (5 nm)/NiFe (4 nm) sample as a function of current density J_P ranging from $1.25 \times 10^{11} \text{ A/m}^2$ to $8.4 \times 10^{11} \text{ A/m}^2$. Figure R1a shows the results that R_{xx} increases quadratically with J_P , measured using DC current (blue squares) and the pulse delta mode (green triangles). Here, the pulse width is 50 μs for the delta mode measurement and a DC current of 0.1s is applied for each current level as indicated in the Inset. Note that the R_{xx} versus J_P measured previously using an oscilloscope is plotted with the black line, which demonstrates that the R_{xx} versus J_P curves are almost the same irrespective of the measurement techniques. Then, we measure the temperature dependence of R_{xx} (Fig. R1b). By comparing those two graphs (Figs. R1a,b), we estimate the sample temperature (T_{sample}) according to J_P as shown in Figure R1c. This indicates that the sample temperature increases to 354 K when a critical switching current of $7.4 \times 10^{11} \text{ A/m}^2$ is applied (Fig. 3b of the original manuscript), consistent with the previous estimation. This is lower than the blocking temperature of the sample (385 K), indicating that the current-induced manipulation of the exchange bias in our experiments is not mainly due to Joule heating.

Next, we examine the substrate effect on Joule heating by comparing the longitudinal resistance R_{xx} of the IrMn (5 nm)/NiFe (4 nm) samples grown on a Si/SiO₂ (200 nm) substrate. Figure R1a shows the increase in R_{xx} of the sample on the Si/SiO₂ substrate (red circles), which

is much faster than the same structured sample on the Si substrate. The large temperature rise in the sample with the Si/SiO₂ substrate is due to the low thermal conductivity of SiO₂ (200 nm), consistent with the reviewer's explanation.

We added the above discussion about the temperature increase due to Joule heating including Fig.R1 (Fig. 2a-c of the main text) in a new section entitled "Current-induced thermal effects" of the revised manuscript.

Figure R1 | Joule heating effects in the IrMn (5 nm)/NiFe (4 nm) sample. **a**, Measurement of longitudinal resistance R_{xx} as a function of current density J_P using DC current (Blue squares), the pulse delta method (green triangles), and using an oscilloscope (black line) in the IrMn/NiFe sample with a Si (675 μm) substrate. Here, a current pulse J_P ranges from $1.25 \times 10^{11} \text{A/m}^2$ to $8.4 \times 10^{11} \text{A/m}^2$. Red circles indicate the R_{xx} vs J_P of the sample grown on a Si/SiO₂ (200 nm) substrate. Inset shows the sequence of the application of DC current. **b**, R_{xx} of the IrMn/NiFe sample measured as a function of temperature, **c**, Estimated sample temperature T_{sample} as a function of J_P . The quadratic line represents a fitting curve of $T=300+98.6J_P^2$.

2.- Describe the substrate in the main text, in particular the thickness of SiO₂ if it is a Si/SiO₂ substrate. While describing the figure of heating (see previous point) give values of resistivity of the layers, although the R_{xx} vs. Temperature (being R_{xx} the resistance of the strip) is perhaps the most valuable information in case someone wants to simulate it. If there is any capping layer that protects the devices (i.e. A 3nm TaOx layer deposited over the entire sample), that should be mentioned too.

Response) We note that we used high-resistive Si substrates without SiO₂ on top and a capping layer of MgO (3.2 nm)/Ta (2 nm) to protect samples from oxidation, which were already

mentioned in the Method section of the revised manuscript. We believe that the mild temperature increase of our sample is due to the large thermal conductivity of the high-resistive Si substrate. To confirm this argument, we measured the thermal conductivities of the Si and Si/SiO₂ (200 nm) substrates. Note that thermal conductivity of a material is given by $\kappa = \rho C_p D$, where ρ , C_p , and D is the bulk density, the specific heat capacity, and the thermal diffusivity, respectively. We first measured the temperature-dependent thermal diffusivity (D) of the substrates using the laser flash method (LFA 457, Netzsch) [*Thermochim. Acta* 455, 46 (2007)]. Figure R2a shows a gradual decrease in D as the temperature increases from 300 K to 400 K, which is attributed to increased phonon–phonon scattering at high temperatures [*Phys. Rev. Appl.* 13, 034011 (2020)]. Then, we measured the specific heat capacity (C_p) of the substrates by differential scanning calorimetry (DSC 214 Polyma, Netzsch) as shown in Fig. R2b. We finally obtained the thermal conductivities (κ) of the Si and Si/SiO₂ substrates (Fig. R2c) by considering the measured D , C_p (Fig. R2a,b), and ρ of 2.3g/cm³ determined by Archimedes method. This shows that the κ of the Si substrate is $\sim 116.5 \text{ Wm}^{-1}\text{K}^{-1}$, which is in good agreement with the literature value of $124 \text{ Wm}^{-1}\text{K}^{-1}$ [*Materials handbook*, 2nd ed.; Springer: London, pp. 465 (2008)]. On the other hand, the Si/SiO₂ substrate has much smaller κ values than the Si substrate for all measured temperatures. This is consistent with the result in Fig. R1a, where a large temperature rise occurs in the sample with the Si/SiO₂ substrate with low thermal conductivity.

We added the above discussion about the thermal conductivities of the substrates in Supplementary Note 6 of the revised manuscript.

Figure R2 | Temperature-dependent thermal conductivities of the substrate. a-c, Temperature-dependent thermal properties of the Si and Si/SiO₂ (200 nm) substrates: thermal diffusivities (a), specific heat capacities (b), and thermal conductivities (c).

3.- The Oersted field generated by the current density was calculated referring to an article *Nat.Nanotech.*12, 980-986 (2017), but at a glance I could not see the link. Ideally, they should calculate the field with a finite element simulation (i.e. COMSOL or similar). If not, provide the formula they are using or refer to it clearly (i.e. formula 3 or Ref X). Then Figure R4d should be re-done. The blue curve in that figure, has to be measured at the temperature they obtain with their thermal estimation and then we will be able to see how far does 1mT go at a larger temperature. This revised figure R4d, has to be brought to the main text, together with the figures described in point 1. It would be a figure dedicated to rule out Joule heating plus Oersted field current annealing.

Response) We feel sorry for not explicitly mentioning the formula used to calculate the Oersted field. We used Eqs. S8-S12 in Supplementary Information 6 of the reference [*Nat. Nanotech.* 12, 980-986 (2017)]. The calculated Oersted field along the y direction ($B^{\text{Oe},y}$) induced in the middle of the NiFe layer of the samples is ~0.68 mT in the IrMn (5 nm)/NiFe (4 nm) sample when applying a switching current density of $7.4 \times 10^{11} \text{ A/m}^2$. Following the reviewer's suggestion, we measured again the planar Hall resistance R_{H} at an elevated temperature, where the current-induced switching occurs. Figure R3 shows R_{H} of the IrMn (5 nm)/NiFe (4 nm) measured at 354 K while rotating a magnetic field of 1 mT, demonstrating that the R_{H} remains almost constant as compared to that measured with 200 mT. This indicates that an Oersted field of less than 1 mT cannot make any noticeable change in the magnetization direction of the exchanged coupled IrMn/NiFe sample, even at the elevated temperature. This result allows us to rule out the current-induced thermal annealing of the IrMn/NiFe as a possible cause of the current-induced manipulation of the exchange bias. Note that there is a minor *sin* component of R_{H} as shown in the Inset, which is probably due to the polar angle induced by the slight misalignment of the sample.

We added the above discussion about the Oersted field effect including Fig. R3 (Fig. 2f of the main text) on page 8 of the revised manuscript.

Figure R3 | Planar Hall resistance R_H of the IrMn/NiFe sample measured with external magnetic fields of 200 mT and 1 mT at 354 K. Inset shows a magnified R_H curve measured at 1 mT.

4.- Line 67. The authors mention again that the anisotropy of the IrMn changes with thickness. How do they know that? The origin of the anisotropy is in the structure of the IrMn and it will unlikely change from one thickness to another if the deposition conditions are the same. Do they mean exchange bias energy?

Response) We appreciate the reviewer's comment. We mentioned that the AFM anisotropy energy (not anisotropy) increases with its thickness. Assuming the anisotropy is constant regardless of the thickness as the reviewer mentioned, the anisotropy energy will increase with its thickness because it is determined by the product of the anisotropy constant and the AFM volume.

Reviewers' Comments:

Reviewer #3:

Remarks to the Author:

The authors have supplied all the information I requested, mainly on the thermal contribution to the effect. I am happy that all that information is now in the main text and the reader can evaluate the thermal contribution. Therefore I have no more hesitations to recommend the publication of the manuscript. While reading, I found some very minor points that the authors may consider.

-Line 67. There is no direct evidence that the torque is applied to the uncompensated moments. Therefore I suggest a less affirmative sentence, something like "Therefore, the SOT seems to be acting on the uncompensated AFM...." or something like that.

-Line 82. Perhaps remove "causing" and substitute the phrase by "pushing the AFM/FM magnetic moment towards the y -direction, parallel to Σ ".

-Line 153. Perhaps rephrase. "This demonstrates that the temperature of the sample in our experiments is lower than the blocking temperature and Joule Heating cannot be the only (or main) cause of the reported effect"

-Line 155: "...switching behaviour is obtained when the experiment is performed at low temperature..."

-Line 160. The phrase starting with "Note that the ...", perhaps should be before the previous phrase. It sounds better to speak first about the estimation of the field and then speak about the consequences that this field has.

Dear reviewers

We appreciate your comments and valuable queries, which have helped us improve the clarity and quality of our manuscript. Given below are detailed point-by-point responses to your questions and suggestions. The corresponding modifications are incorporated in the revised manuscript (marked in blue). We believe our manuscript has been improved significantly and now deserves publication in *Nature Communications*.

Yours sincerely,

Byong-Guk Park on behalf of all co-authors

[Reviewer #3]

The authors have supplied all the information I requested, mainly on the thermal contribution to the effect. I am happy that all that information is now in the main text and the reader can evaluate the thermal contribution. Therefore, I have no more hesitations to recommend the publication of the manuscript. While reading, I found some very minor points that the authors may consider.

Response) We thank the reviewer for carefully reading our responses. We are happy to hear that *“Therefore, I have no more hesitations to recommend the publication of the manuscript.”* Following the reviewer’s suggestions, we revised the manuscript as described below.

-Line 67. There is no direct evidence that the torque is applied to the uncompensated moments. Therefore I suggest a less affirmative sentence, something like “Therefore, the SOT seems to be acting on the uncompensated AFM...”... or something like that.

Response) Following the reviewer’s suggestion, we revised the sentence to be less affirmative as *“therefore, the SOT seems to be applied to the uncompensated AFM moments at the IrMn/NiFe interface,”* on page 4 of the revised manuscript.

-Line 82. Perhaps remove “causing” and substitute the phrase by “pushing the AFM/FM magnetic moment towards the y-direction, parallel to Sigma”.

Response) Following the reviewer’s suggestion, we rephrased the word *“causing”* to *“pushing the AFM/FM magnetic moment towards the y-direction, parallel to σ ”* on page 4 of the revised manuscript.

-Line 153. Perhaps rephrase. “This demonstrates that the temperature of the sample in our experiments is lower than the blocking temperature and Joule Heating cannot be the only (or main) cause of the reported effect”

Response) Following the reviewer’s suggestion, we revised the sentence as *“This demonstrates that the temperature of the sample in our experiments is lower than the blocking temperature and Joule heating cannot be the main cause of the observed effect”* on page 8 of the revised manuscript.

-Line 155: "...switching behaviour is obtained when the experiment is performed at low temperature..."

Response) Following the reviewer's suggestion, we revised the sentence as "*Furthermore, we observed that a similar switching behavior is obtained when the experiment is performed at low temperatures down to 100 K*" on page 8 of the revised manuscript.

-Line 160. The phrase starting with "Note that the ...", perhaps should be before the previous phrase. I sounds better to speak first about the estimation of the field and then speak about the consequences that this field has.

Response) Following the reviewer's comment, we revised the phrase as "*To this end, we first calculated Oersted field acting on the NiFe layer in the IrMn/NiFe sample (Supplementary Note 9), which is ~ 0.68 mT at a switching current density of 7.4×10^{11} A/m²,*" and placed it before the sentence describing the consequences that this field has on page 8 of the revised manuscript.